# Nonequilibrium ordering dynamics of confined soft alginate hydrogel colloids driven by time-evolving electrostatic interactions

In Hwan Jung[1,5], Chetan C. Revadekar[1,5], Hag Sung Lee[1], Hyerim Hwang [2,3] ✉, Hyosung An [4] ✉ & Bum Jun Park [1] ✉

Elucidating how repulsive interactions evolve to generate ordered structures in nonequilibrium colloidal systems remains a central challenge, partly because few experimental platforms provide particle-resolved access to their structural evolution. Here we show that alginate hydrogel colloids confined within cyclohexyl bromide (CHB) emulsion droplets form a controllable model system in which electrostatic interactions evolve in time and drive ordering. $Ba^{2+}$ ions diffusing from the surrounding aqueous phase progressively cross-link the alginate droplets, increasing their surface charge, while buoyancy compacts them into locally quasi-two-dimensional layers within the CHB phase. As electrostatic repulsion strengthens, the assembly evolves from a disordered state to a hexagonally ordered structure. By calibrating Brownian dynamics simulations to the experimentally measured lattice spring constant, we constrain the effective Debye screening length to ≈2.5−3 μm. Quantitative imaging further shows that ordering emerges once a dimensionless interaction parameter—defined as the ratio of electrostatic interaction energy to thermal energy—reaches values of ≈117−149. The ordered state exhibits reversible disordered−order behavior under mechanical and magnetic perturbations, demonstrating a robust nonequilibrium platform for probing charge-regulated colloidal ordering under confinement.

Colloidal systems provide a versatile model platform for studying interaction-driven ordering at experimentally accessible length and time scales. Particle-resolved imaging, tunable interactions, and slow relaxation dynamics enable real-time visualization of ordering pathways, offering insights that complement those from atomic and electronic systems[1–7]. In low-dielectric media such as CHB–decalin mixtures, long-range screened electrostatic repulsion under weak screening can stabilize ordered colloidal phases[8–10]. Such systems exhibit rich behaviors, including re-entrant melting, glass formation, and tunable elasticity, which are well described by established theoretical frameworks such as Yukawa, one-component-plasma, and charge-regulation models[11,12]. Despite these advances, most prior

[1]Department of Chemical Engineering (BK21 FOUR Integrated Engineering Program), College of Engineering, Kyung Hee University, Yongin, Gyeonggi-do, South Korea. [2]Department of Chemical Engineering and Materials Science, Ewha Womans University, Seodaemun-gu, Seoul, South Korea. [3]Institute for Multiscale Matter and Systems (IMMS), Ewha Womans University, Seoul, Republic of Korea. [4]Department of Petrochemical Materials Engineering, Chonnam National University, Yeosu, Jeollanam-do, South Korea. [5]These authors contributed equally: In Hwan Jung, Chetan C. Revadekar. ✉e-mail: hyerimhwang@ewha.ac.kr; hyosungan@jnu.ac.kr; bjpark@khu.ac.kr

studies have focused on equilibrium structures, whereas the time-dependent mechanisms governing surface-charge buildup, evolving screening length, and ordering under confinement have not been extensively explored.

Interaction-driven ordering in colloidal systems generally arises when electrostatic interactions become comparable to or exceed thermal fluctuations, leading to the emergence of ordered structures. In the soft-matter literature, this regime is sometimes described using the term Wigner-type or Wigner-like ordering[9–12], reflecting its historical origin in the concept of Wigner crystallization, in which ordered electronic states emerge from the competition between long-range Coulomb interactions and kinetic energy at low carrier densities[13–17]. In the colloidal assemblies, this terminology is used to denote a regime in which electrostatic interactions dominate over thermal fluctuations and give rise to ordered structures. In the present work, we focus on this interaction-dominated ordering behavior in colloidal particles under screened electrostatic interactions.

Here, we develop a room-temperature platform for studying time-dependent nonequilibrium ordering using soft alginate hydrogel colloids confined within CHB droplets dispersed in an aqueous phase. A microfluidic device produces alginate-nested emulsions with well-controlled droplet size and interfacial curvature, providing reproducible geometric confinement for monitoring ordering pathways. CHB, with its low-dielectric constant, supports strong electrostatic repulsion under weak screening[10,18], while divalent cations (e.g., Ba²⁺) diffusing from the aqueous phase progressively crosslink the alginate droplets, gradually increasing their surface charge. Buoyancy-driven stratification then yields quasi-two-dimensional (2D) layers that spontaneously develop hexagonal order. As a result of this ion-mediated charging process, the strength of interparticle electrostatic interactions evolves continuously in time. In this context, nonequilibrium refers to the temporal evolution of interparticle interaction strength.

Although local particle configurations can relax into quasi-steady structural states on relatively short timescales at a given interaction strength, the electrostatic interaction strength itself evolves more slowly due to progressive charge buildup. As a result, the global emergence of long-range hexagonal order is governed by this slower timescale, rendering the overall ordering process intrinsically none-quilibrium. We quantify the onset and kinetics of this ordering using particle-resolved imaging, Voronoi/Delaunay analyses, and fast Fourier transform (FFT)–based metrics. To establish a quantitative interaction scale, we calibrate Brownian dynamics (BD) simulations to the experimentally measured lattice spring constant, enabling estimation of the Debye screening length under our experimental conditions. We then characterize the time-dependent strengthening of electrostatic interactions using a dimensionless interaction parameter $\Gamma$, defined as the ratio of electrostatic energy to thermal energy. We further map layer-resolved structural ordering and examine the effects of ion valence and ion concentration. Finally, we demonstrate reversible disorder-order transitions under magnetic and mechanical perturbations, together establishing a controllable nonequilibrium platform for studying classical, interaction-driven colloidal ordering.

## Results
### Self-organization of alginate colloids
We fabricated a hierarchically nested emulsion by co-flow microfluidics[19–23], generating alginate-in-CHB (Alg@CHB) droplets in an aqueous continuous phase (Fig. 1a, b). The inner phase comprised sodium alginate droplets dispersed in CHB (dielectric constant $\varepsilon \approx 7.9$), which were prepared by bath sonication of an alginate aqueous solution mixed with CHB, resulting in inherently polydisperse alginate sub-droplets. The outer phase was ultrapure water containing 2 wt.% poly(vinyl alcohol) (PVA) as a stabilizer. The resulting Alg@CHB emulsions with a monodisperse size (mean diameter ≈400 μm) were

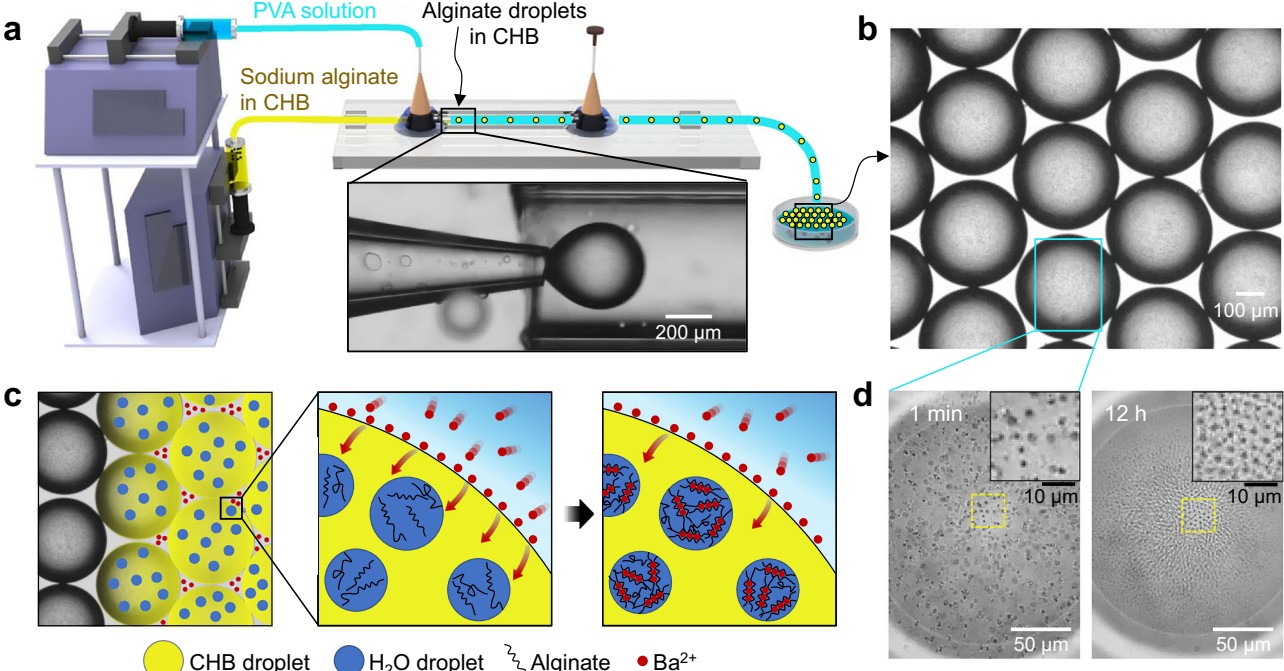

**Fig. 1 | Nested emulsion platform for Ba²⁺-induced alginate hydrogelation.**
**a** Microfluidic production of CHB droplets encapsulating numerous aqueous alginate sub-droplets (Alg@CHB) in an outer 2 wt.% PVA solution (schematic; not to scale). Inset shows a bright-field image of CHB droplet formation. **b** A bright-field image of monodisperse Alg@CHB emulsions collected in water containing 40 mM (≈1 wt.%) Ba(Ac)₂. **c** Schematic of hydrogelation: Ba²⁺ ions from the external aqueous phase cross the CHB/aqueous interface and reach the internal alginate aqueous sub-droplets, where they ionically crosslink alginate to form hydrogel particles (not to scale). **d** Time-lapse bright-field images showing the emergence and structural evolution of alginate hydrogel particles within a single CHB droplet at 1 min and 12 h after Ba²⁺ exposure. Insets show magnifications of the yellow boxed regions.

collected and incubated in an aqueous barium acetate solution ($C_{Ba(Ac)_2}$ = 40 mM ≈1 wt.%), under which conditions ionically cross-linked alginate hydrogels formed, improving particle stability.

Alginate crosslinking occurred within the CHB droplets (Fig. 1c). Divalent $Ba^{2+}$ ions accumulated at the water/CHB interface and diffused through the CHB phase to the internal alginate sub-droplets, where ion exchange at carboxylate groups replaced $Na^+$ and formed $Ba^{2+}$-mediated ionic crosslinks, yielding soft hydrogel particles[24–28]. Charged colloidal particles in CHB have been reported to experience unusually strong electrostatic repulsion owing to CHB's low dielectric screening[10,18]. Within each CHB emulsion droplet, buoyancy drove floatation-induced stratification of alginate hydrogel particles, producing a stack of locally quasi-2D layers that self-organized into a hexagonally ordered lattice over hours (Fig. 1d). Control emulsions lacking alginate (water-only sub-droplets in CHB) showed no detectable ordering under otherwise identical conditions (Supplementary Fig. 1), suggesting that inter-sub-droplet repulsion is insufficient in the absence of $Ba^{2+}$-mediated alginate crosslinking.

The time dependence of interparticle repulsion was corroborated by zeta-potential measurements in a vial geometry with a $Ba(Ac)_2$ aqueous layer over an alginate-in-CHB dispersion (Supplementary Fig. 2). The magnitude of zeta potential $\zeta$ increased with incubation time, indicating progressive surface-charge buildup from $Ba^{2+}$–carboxylate interactions and continued crosslinking (Supplementary Note 1), consistent with the observed ordering kinetics (Fig. 1d). Although the vial geometry does not replicate the confined droplet environment exactly, it captures the same $Ba^{2+}$-mediated charging mechanism in the CHB phase and thus provides a reasonable effective measure of the evolving interaction strength.

### Time-dependent evolution of alginate colloidal ordering

We reconstructed the internal structure of an Alg@CHB droplet using z-stack optical microscopy and generated a 3D rendering of particle coordinates after 12 h in $Ba^{2+}$ solution (Fig. 2a, b). Within the CHB phase, larger alginate particles preferentially occupied higher elevations, whereas smaller ones resided deeper in the droplet, consistent with size-dependent flotation (larger particles attain higher terminal rise velocities). An order-of-magnitude estimate based on Stokes' law indicates that droplets in the ≈0.5–5 µm size range traverse a 400 µm CHB droplet within ~$10^2$–$10^4$ s (minutes to a few hours; Supplementary Note 2). Particles near the top therefore form quasi-2D layers on relatively short timescales, providing a geometrically confined environment for subsequent ordering. A schematic of the formation pathway is provided in Supplementary Fig. 3. Overall, the ordering arises under nonequilibrium conditions from the coupled evolution of buoyancy-driven stratification and surface-charge buildup during $Ba^{2+}$-mediated hydrogelation.

Figure 2c shows top-layer images at 1, 4, 6, 8, and 12 h, revealing a progression from a loosely correlated arrangement to a pronounced hexagonally ordered lattice. Two concurrent processes govern this evolution: (i) an increase in particle surface charge with incubation time (Supplementary Fig. 2) and (ii) flotation-driven, size-selective enrichment of larger particles in the topmost layer. For quantification, particle centroids were detected and used to compute Delaunay triangulations and Voronoi tessellations (Fig. 2d and Supplementary Fig. 4). In the top layer, the mean particle diameter increased from $\langle D \rangle$ ≈ 3.76 to 4.91 µm over 12 h (Fig. 2e), reflecting the progressive composition change of that layer toward larger particles. The mean center-to-center spacing increased from $\langle r_{cc} \rangle$ ≈ 9.95 to 12.62 µm, attributable to stronger electrostatic repulsion driven by both surface-charge buildup and size-selective stratification that enriches larger particles in the top layer. Despite the increase in $\langle D \rangle$, the 2D packing fraction ($\phi$) decreased, indicating that repulsion-driven expansion of interparticle spacing outpaced the diameter increase and/or reduced

the local number density. The standard deviation of $r_{cc}$ narrowed from ≈3.09 to ≈1.85 µm (Fig. 2e), indicating enhanced positional order.

To visualize local symmetry, we analyzed nearest-neighbor displacement vectors $|\Delta \mathbf{r}| = (\Delta x, \Delta y)$ extracted from the Delaunay meshes (Fig. 2f). At $t$ = 1 h, points were broadly scattered ($\langle r_{cc} \rangle$ = 9.95 ± 3.09 µm), consistent with weak short-range order. By $t$ = 4–6 h, the distribution collapsed onto a ring at $|\Delta \mathbf{r}| \approx \langle r_{cc} \rangle$, reflecting a better defined nearest-neighbor shell. By $t$ = 8–12 h, the ring showed strong azimuthal modulation with ≈60° spacing, characteristic of a hexagonally ordered lattice. Because the horizontal axis uses $|\Delta x|$ (left-right folding), the sixfold pattern collapses to at most four high-density lobes in the map.

Voronoi analyses of the top layer (Fig. 2g) further captured the growth of hexagonal order. The mean coordination number remained near six (5.6 ± 1.2, 5.7 ± 1.5, 5.8 ± 0.9, 5.9 ± 0.8, and 5.9 ± 0.6 at $t$ = 1, 4, 6, 8 and 12 h, respectively), while the distribution narrowed in time (Fig. 2h), evidencing a rising fraction of six-coordinated cells. Residual five- and seven-fold sites at 12 h likely reflect size dispersity ($\langle D \rangle$ = 4.92 ± 0.78 µm) and associated topological defects. Together, these trends demonstrate a gradual, nonequilibrium ordering from a weakly correlated layer to a hexagonally ordered colloidal state.

### Layer-specific analysis of alginate colloidal ordering

To analyze the structure layer by layer in 2D, we acquired z-stacks with an axial step of $\Delta z$ = 1.1 µm and segmented the colloidal system into three layers. The particle count along z showed distinct peaks (Fig. 3a), defining the top layer L1, and two deeper layers, L2 and L3, located 10.8 and 17.0 µm below L1, respectively. For each layer we detected particle centroids, computed Delaunay triangulations and Voronoi tessellations, and evaluated FFT patterns (Fig. 3b).

A clear order gradient emerged with depth. In the centroid maps (Fig. 3b, second column) positions became increasingly irregular from L1 to L3. Voronoi analysis yielded mean coordination numbers of 6.0 ± 0.1 (L1), 6.0 ± 0.4 (L2), and 5.9 ± 0.9 (L3), indicating that L1 is nearly perfectly sixfold coordinated whereas L3 contains more five- and seven-coordinated defects. Consistently, FFT showed a sharp hexagonal spot pattern for L1, a weaker pattern for L2, and a diffuse ring for L3, reflecting a progressive reduction in hexagonal order. Delaunay overlays (Supplementary Fig. 5) and neighbor-displacement maps ($\Delta x$, $\Delta y$) (Fig. 3c) corroborated this trend: L1 displayed a well-defined annulus with pronounced angular clustering, L2 retained the annulus but with more diffuse angular clustering, and L3 showed a largely dispersed cloud.

Layer-resolved metrics (Fig. 3d) highlights the governing factors. The mean particle diameter decreased from $\langle D \rangle$ = 4.4 ± 0.7 µm in L1 to 2.7 ± 0.3 µm in L3, consistent with size-selective flotation that enriches larger particles near the top. The mean center-to-center spacing decreased slightly from $\langle r_{cc} \rangle$ = 14.4 ± 2.1 µm (L1) to 13.0 ± 5.3 µm (L3), but its standard deviation increased, indicating growing position disorder. The 2D packing fraction $\phi$ likewise declined with depth, implying weaker effective repulsion and a progressive loss of hexagonal order. Taken together, the layer-by-layer analysis shows that the colloidal assembly is most strongly ordered in the upper layer and becomes progressively less ordered with depth, a consequence of size-selective stratification and the associated reduction in interparticle repulsion in deeper layers.

### Ion-dependent alginate colloidal ordering

$Ba(Ac)_2$ concentration strongly affected ordering, with a sharp change between 0 and 0.5 mM (Fig. 4). Alg@CHB droplets were incubated in $C_{Ba(Ac)_2}$ = 0–100 mM and analyzed after 12 h (top layer, L1). At 0 mM, droplets coalesced (no crosslinking) and no lattice formed. At 0.5 mM, hexagonal domains appeared (Fig. 4a, b). Increasing $C_{Ba(Ac)_2}$ to ≥ 50 mM—consistent with observations at 40 mM in Figs. 2 and 3—further suppressed residual coalescence: the neighbor-displacement

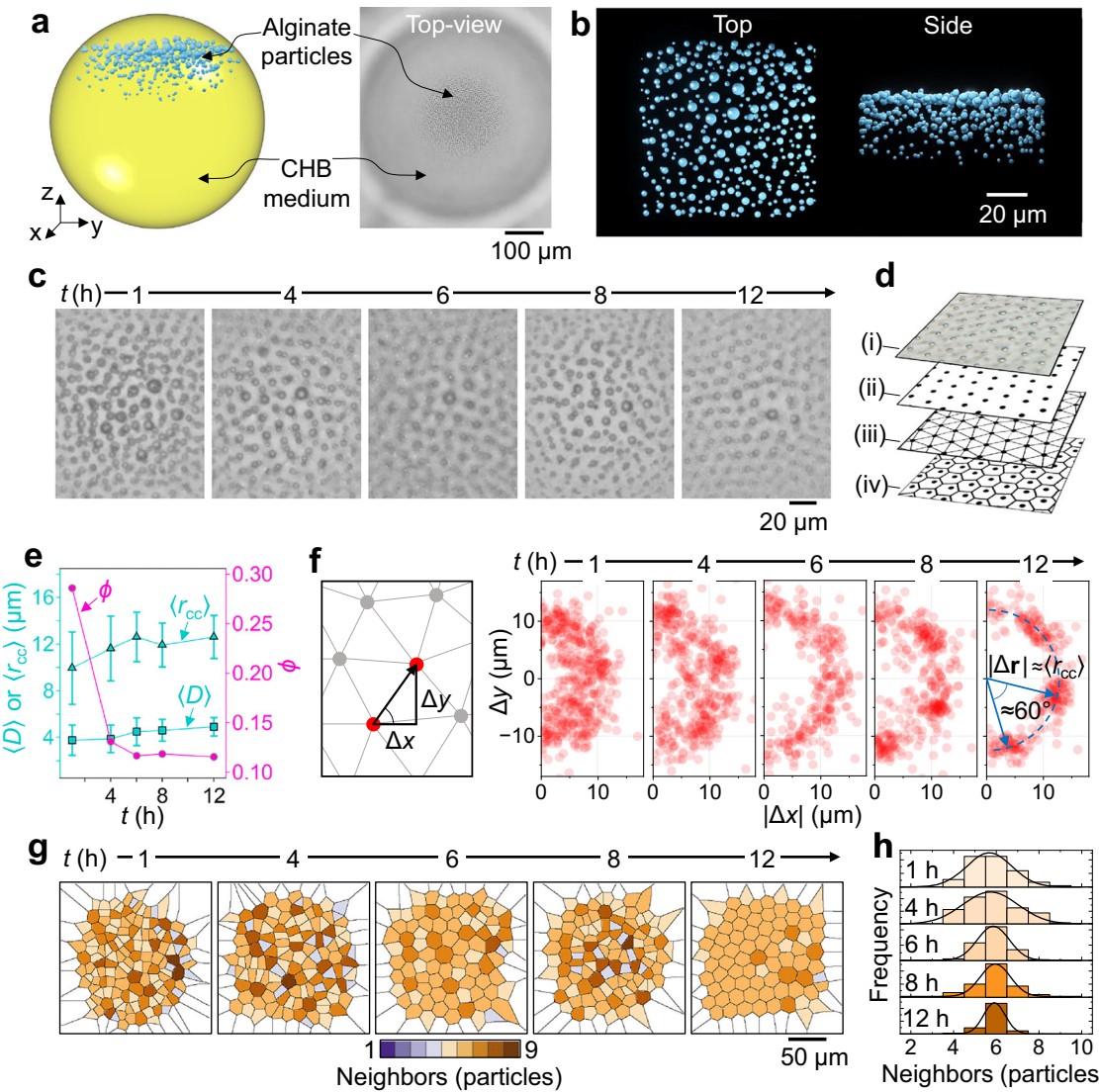

**Fig. 2 | Time-resolved self-organization and layering of alginate colloidal ordering inside CHB droplets. a** Schematic and top-view bright-field image showing alginate particles concentrating within the CHB droplet, where they assemble into locally quasi-2D ordered domains (12 h in 40 mM Ba(Ac)$_2$). **b** 3D rendering from $z$-stack particle coordinates, displayed top and side views, revealing multilayer stacking. **c** Top-layer bright-field images at 1, 4, 6, 8, and 12 h illustrating progressive ordering. **d** Workflow for alginate colloidal structure analysis: raw image (i), particle centroids (ii), Delaunay triangulation (iii), and Voronoi tessellation (iv). **e** Temporal evolution of layer-resolved structural metrics ($\langle D \rangle$, $\langle r_{cc} \rangle$, and $\phi$) for the top-layer. Error bars represent the standard deviations derived from ≥64 particles for $\langle D \rangle$ and ≥309 data points for $\langle r_{cc} \rangle$. **f** Neighbor-displacement maps ($\Delta x$, $\Delta y$) at each time point. **g** Voronoi maps of the top layer color-coded by coordination number (number of neighbors). **h** Histograms of coordination number for the top layer at each time point. Source data are provided as a Source Data file.

maps showed a clear annulus with pronounced angular clustering (Fig. 4c), and the Voronoi analysis indicated a higher fraction of sixfold cells (Fig. 4d). In general, for $C_{Ba(Ac)_2} \gtrsim 40$ mM and $t \gtrsim 8$ h, structural metrics changed only modestly, indicating that crosslinking and electrostatic repulsion were established; the lattice thus likely entered a quasi-steady structural state under nonequilibrium conditions, in which further increases in concentration or incubation time had little additional effect on ordering.

Ion identity also played a key role (Supplementary Fig. 6a). In 40 mM solutions of monovalent ions (K$^+$, Na$^+$), alginate sub-droplets remained largely random, consistent with weak hydrogelation and insufficient repulsion. In contrast, divalent ions (Ca$^{2+}$, Ba$^{2+}$) produced hexagonally ordered lattices with relatively uniform spacing. Ba$^{2+}$ yielded the most stable ordered state: barium-alginate lattices persisted to 48 h, whereas calcium-alginate lattices gradually degraded. This trend accords with reports that Ba$^{2+}$ forms stronger alginate crosslinks than other common cations due to its higher charge density and stronger binding to alginate carboxylates[27]. In line with this, Ba$^{2+}$ more effectively promotes interfacial charging, thereby strengthening repulsive interactions and stabilizing the lattice.

## Estimation of Debye screening length

To estimate the Debye screening length $\kappa^{-1}$, we calibrated BD simulations to the experimentally measured lattice spring constant ($k_s^{exp}$) extracted from cage-relative thermal displacements (Fig. 5a and Supplementary Movie 1)[29–31]. The spring constant estimator was independently validated by simulating a single particle in a known 2D harmonic trap and recovering the input stiffness (see Supplementary Fig. 7). Here, the spring constant serves as a quantitative descriptor of the local stiffness experienced by each particle within the ordered colloidal lattice, reflecting the degree of electrostatic confinement imposed by neighboring particles[32].

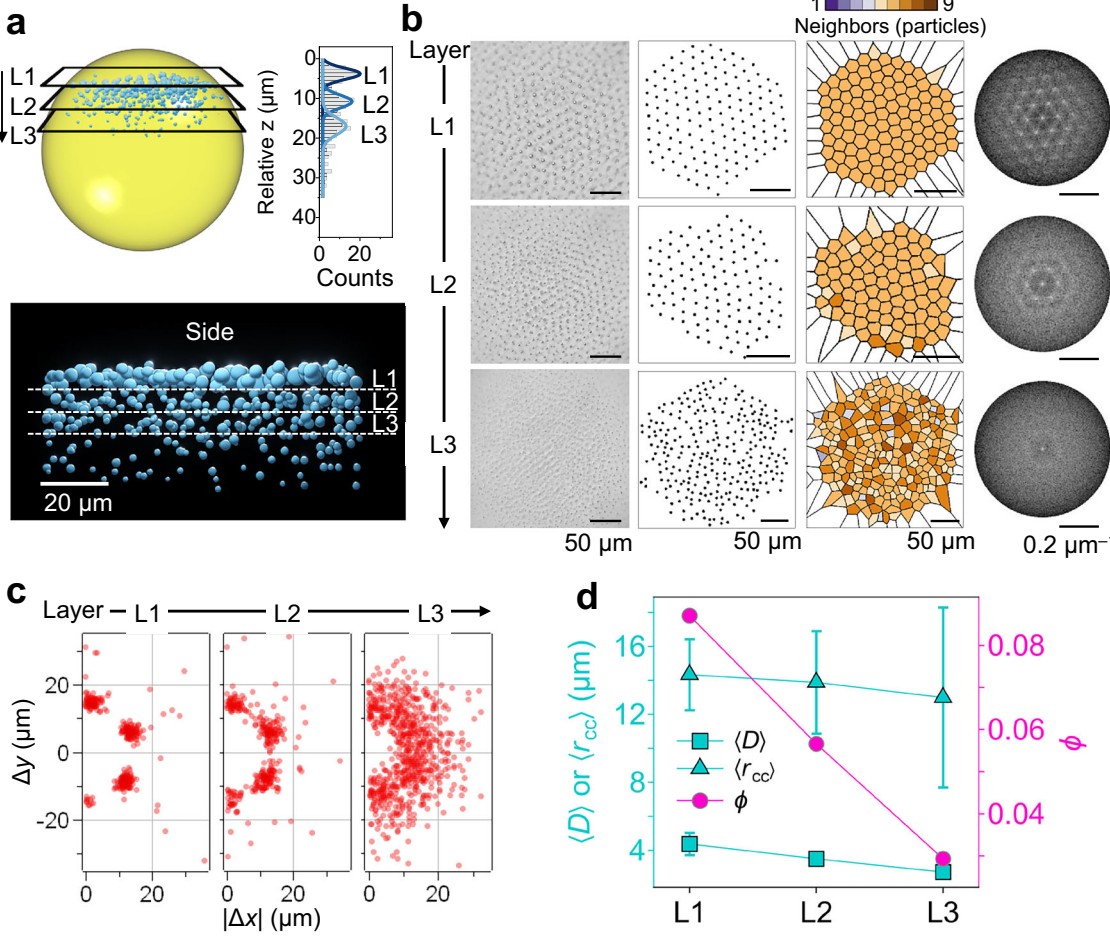

**Fig. 3 | Layer-resolved structure of multilayered colloidal ordering. a** Schematic (top left) and 3D side-view (bottom) rendering from z-stack microscopy showing stratified layers within the CHB phase after 12 h in 40 mM Ba(Ac)$_2$. The particle count versus relative z (top right) identifies three layers (L1–L3). **b** For each layer (top to bottom: L1, L2, L3): bright-field image; centroid map; Voronoi tessellation color-coded by coordination number; and FFT pattern. **c** Neighbor-displacement maps ($\Delta x$, $\Delta y$) from Delaunay meshes for each layer (Supplementary Fig. 5). **d** Layer-resolved structural metrics ($\langle D \rangle$, $\langle r_{cc} \rangle$, and $\phi$) for each layer. Error bars represent the standard deviations derived from ≥90 particles for $\langle D \rangle$ and ≥337 data points for $\langle r_{cc} \rangle$. Source data are provided as a Source Data file.

Using a Yukawa potential (Supplementary Fig. 8), we performed BD simulations in which both the zeta potential $\zeta$ and the screening length $\kappa^{-1}$ were systematically and independently varied to obtain the lattice spring constant $k_s^{BD}(\zeta, \kappa^{-1})$ (Fig. 5b). Comparison of this simulated parameter space with the experimentally measured $k_s^{exp}$ yields a mismatch map, $\Delta k_s = |k_s^{BD} - k_s^{exp}|$ (Fig. 5c), from which the best agreement ($|\Delta k_s| \approx 0$) is identified. The corresponding region (pink shading in Fig. 5b, c) lies within an experimentally reasonable range of $\zeta \approx 50$–60 mV (Supplementary Fig. 2) and yields an effective Debye screening length of $\kappa^{-1} \approx 2.5$–3 µm.

Because the ordering process evolves under conditions of continuously changing interaction strength, assigning unique values of $\zeta$ and $\kappa^{-1}$ at any specific time is inherently difficult. Nevertheless, by assuming a representative $\zeta \approx 50$ mV, the BD–experiment matching yields an effective screening length of $\kappa^{-1} \approx 3$ µm. This estimate is consistent with our prior optical tweezer measurements on poly(methyl methacrylate) (PMMA) particles in CHB/n-decane mixtures, which reported $\kappa^{-1} \approx 2.5$ µm[18]. More broadly, reported screening lengths in nonpolar or weakly polar media fall in the micrometer range—for example, ≈1 µm for PMMA particles in cis-decalin/CHB with poly-12-hydroxystearic acid[33], ≈ 0.5–12 µm for polystyrene or PMMA particles in hexadecane with cationic surfactant (sodium di-2-ethyl-hexylsulfosuccinate)[34,35], and ≈0.8–8 µm for PMMA in hexane with sorbitan trioleate[36].

In the present system, the CHB phase remains weakly ionic, and the effective screening length $\kappa^{-1}$ in CHB evolves in time as Ba$^{2+}$ diffuses

from the aqueous phase and is progressively consumed by alginate crosslinking. Despite this gradual evolution, the inferred screening length $\kappa^{-1} \approx 3$ µm indicates a trace but finite ionic strength in CHB. Assuming that Ba(Ac)$_2$ provides the dominant mobile ions in CHB while Na$^+$ remains largely bound within the hydrogel network, the corresponding ion concentration can be estimated as $C_{Ba(Ac)_2} = \frac{1}{3}\frac{\varepsilon_0 \varepsilon k_B T}{e^2 N_A (\kappa^{-1})^2} = 6.91 \times 10^{-10}$ M for $\kappa^{-1} \approx 3$ µm. This dilute ionic strength is sufficient to account for the observed electrostatic screening in CHB, supporting our interpretation of electrostatically driven colloidal ordering under weakly screened conditions.

To assess the robustness of the spring-constant–based calibration, we additionally examined how particle-size dispersity (i.e., the coefficient of variation, CV) influences the simulated lattice mechanics. We introduced experimentally relevant size variations by sampling particle radii from a Gamma distribution[37,38] and performed BD simulations at matched surface coverage. Increasing dispersity reproduces the expected melting-like microstructural disorder (Supplementary Fig. 9a–i), consistent with prior studies[37,38], yet the resulting lattice spring constant remains largely insensitive to CV over the range tested (Supplementary Fig. 9j and Supplementary Note 3). This insensitivity indicates that the lattice spring constant predominantly reflects the underlying interaction scale rather than the degree of dispersity, supporting the reliability of the $k_s$-based screening-length calibration used in Fig. 5b, c.

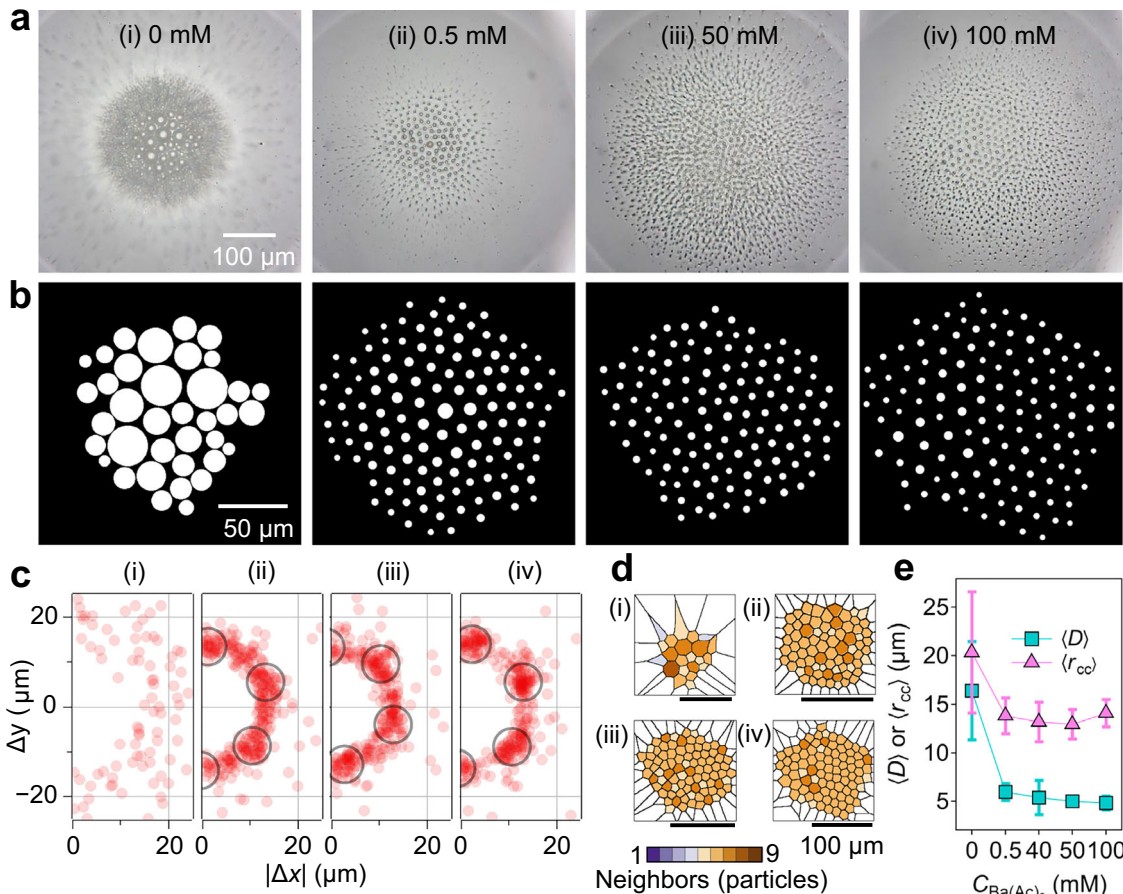

**Fig. 4 | Effect of Ba(Ac)2 concentration on the structure of alginate colloidal ordering (top layer, L1). a** Bright-field images after 12 h showing L1 at $C_{\text{Ba(Ac)}_2} = 0$ (i), 0.5 (ii), 50 (iii), and 100 mM (iv). **b** Corresponding segmented particle maps obtained from the central, in-focus zoomed-in region and used for analysis. **c** Neighbor-displacement maps ($\Delta x$, $\Delta y$) at each concentration; black circles highlight the annular high-probability region at $|\Delta \mathbf{r}| \approx \langle r_{\text{cc}} \rangle$. **d** Voronoi tessellations color-coded by coordination number. **e** Quantitative trends of $\langle D \rangle$ and $\langle r_{\text{cc}} \rangle$ versus $C_{\text{Ba(Ac)}_2}$. Error bars represent the standard deviations derived from ≥37 particles for $\langle D \rangle$ and ≥90 data points for $\langle r_{\text{cc}} \rangle$. Source data are provided as a Source Data file.

## Interaction parameter and time-dependent ordering

We define a dimensionless interaction parameter $\Gamma$ to quantify the relative strength of electrostatic interactions compared to thermal fluctuations in our colloidal system dispersed in CHB. Interparticle repulsion is modeled using the Yukawa (screened Coulomb) potential $E_{\text{pot}} = \frac{4\pi\varepsilon_0 \varepsilon \zeta^2 R^2 e^{2\kappa R}}{r e^{\kappa r}}$, where $R$ is the particle radius, $r$ is the interparticle separation, $\varepsilon_0$ is the vacuum permittivity, and $\kappa^{-1}$ is the Debye screening length. Although the alginate particles are mechanically soft, their interactions in CHB are dominated by the charged surface layer and occur at separations far larger than those at which mechanical deformation would be relevant. The Yukawa potential therefore provides an appropriate effective pair interaction for the present system. Using the thermal energy $k_B T$ as the reference energy scale, the interaction parameter is defined as $\Gamma = \frac{E_{\text{pot}}}{k_B T} = \frac{\pi\varepsilon_0 \varepsilon \zeta^2 \langle D \rangle^2 e^{\kappa \langle D \rangle}}{k_B T \langle r_{\text{cc}} \rangle e^{\kappa \langle r_{\text{cc}} \rangle}}$, using experimentally measured values of $\langle D \rangle$, $\langle r_{\text{cc}} \rangle$, and $\zeta$ (Fig. 2e and Supplementary Fig. 2). Low $\Gamma$ corresponds to weakly correlated, fluid-like configurations, whereas large $\Gamma$ indicates increasingly interaction-dominated, ordered colloidal states.

Figure 5d shows that $\Gamma$ increases with incubation time and with increasing screening length, reflecting the progressive strengthening of electrostatic repulsions. A slight dip in $\Gamma$ at shorter screening lengths can be attributed to the lower $\langle r_{\text{cc}} \rangle$ values observed in the disordered state. Considering the range of $\kappa^{-1} \approx 2.5$–3 µm estimated from the BD analysis (Fig. 5b, c), $\Gamma$ reaches values of ≈117–149 at $t \approx 6$ h, coinciding

with the experimentally observed onset of hexagonal ordering (Fig. 2). At later times ($t \gtrsim 8$ h), $\Gamma$ exceeds ≈220 (pink regions in Fig. 5d), corresponding to a strongly interaction-dominated regime in which robust colloidal ordering is established. Although the microscopic mechanisms underlying electronic Wigner crystallization are fundamentally different from those governing colloidal ordering, the interaction parameter values at which stable ordering emerges in our colloidal system fall within the numerical range often quoted for screened electronic Wigner systems (≈95–150)[14–16,39]. Here, this numerical correspondence is not interpreted as evidence of physical equivalence, but rather as providing a useful reference point for identifying the boundary between weakly correlated and strongly ordered regimes in colloidal assembly systems.

## Reversible ordering under external perturbations

We tested whether the ordered assemblies could recover after external perturbations (Fig. 6a, b). In magnetically responsive samples (alginate solution doped with $Fe_2O_3$ nanoparticles during Alg@CHB preparation; see Methods), a neodymium magnet placed beside the container produced a direction field that drove particles along the field axis, transiently disrupting the ordered structure (Fig. 6a). After removal of the magnetic field, the structure spontaneously reassembled within ≈20 min, as seen by the reappearance of a clear annulus with pronounced angular clustering in the neighbor-displacement map (Fig. 6c). A reversible transition was also observed under gentle mechanical agitation: brief shaking disordered the structure, whereas

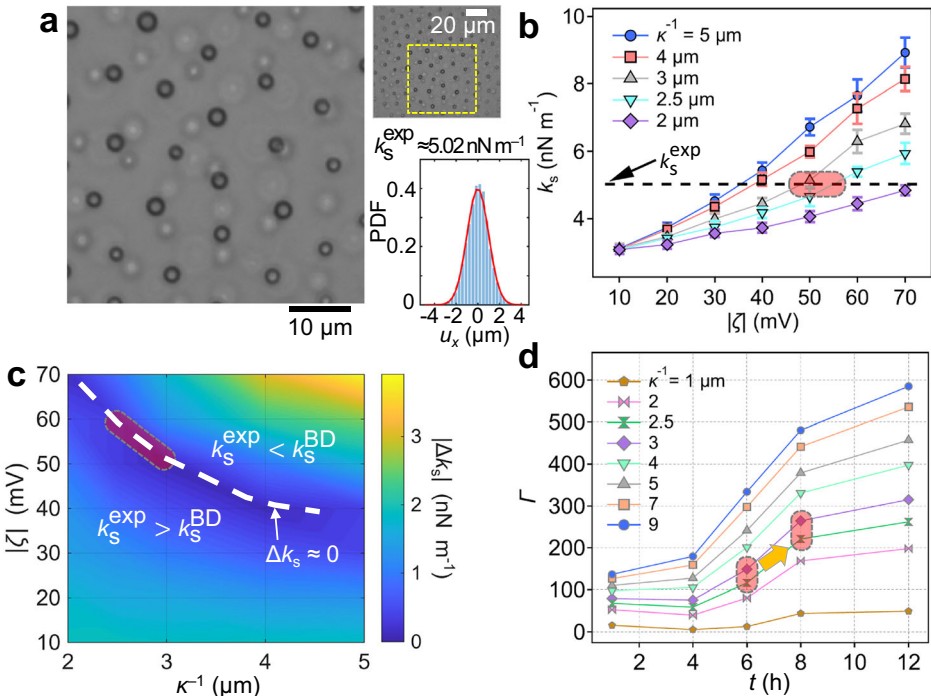

**Fig. 5 | Estimation of the Debye screening length and the dimensionless interaction parameter. a** Bright-field image of an ordered lattice after 12 h in 100 mM Ba(Ac)$_2$, used to extract the experimental spring constant $k_s^{exp}$ (Supplementary Movie 1, recorded at 30 fps). The top-right inset displays the full field of view with the analyzed region (yellow box). The bottom-right inset shows the probability density function (PDF) of the $x$-displacement $u_x$ with a Gaussian fit (solid red line), yielding $k_s^{exp} \approx 5.02$ nN m$^{-1}$. **b** BD-predicted lattice spring constant $k_s^{BD}$ as a function of zeta potential $\zeta$ for various $\kappa^{-1}$. The horizontal dashed line denotes $k_s^{exp}$.

Error bars are standard deviations over ten independent BD runs. **c** Contour map of the mismatch $\Delta k_s = |k_s^{BD} - k_s^{exp}|$ in the $(\zeta, \kappa^{-1})$ plane. The white dashed curve marks $\Delta k_s \approx 0$. **d** Dimensionless interaction parameter $\Gamma$ as a function of incubation time $t$ for different Debye screening lengths $\kappa^{-1}$. The pink regions in panels **b–d** highlight the parameter band most consistent with experiment, constraining the Debye length to the few-micrometer range ($\kappa^{-1} \approx 2.5$–3 µm) and the zeta potential to around $\zeta \approx 50$ mV. Source data are provided as a Source Data file.

cessation of agitation led to gradual reorganization and recovery of the hexagonal ordering signature (Fig. 6b, d). These behaviors indicate that the ordered colloidal assemblies behave as reconfigurable, self-healing nonequilibrium solids, with recovery governed by buoyancy-driven layering and strong electrostatic repulsion.

### Alginate ordering in a flattened sessile drop

In addition to the confined CHB droplets, we verified that alginate colloidal ordering also emerges in a flattened CHB sessile drop geometry with a much larger interfacial radius of curvature (Supplementary Fig. 10a). Despite the altered geometry, the system exhibited a time scale of ≈8 h for ordering, with well-defined hexagonal domains appeared in the CHB phase after exposure to Ba$^{2+}$, consistent with the droplet-based experiments. This demonstrates that the ordering mechanism is robust and not restricted to the confined droplet environment, and that the sessile drop approach effectively illustrates the generality of the phenomenon. These observations also support the use of vial-based $\zeta$-potential measurements to describe the electrostatic charging behavior of alginate hydrogel particles in both geometries.

To clearly visualize the ordering process, we monitored the edge region of the CHB sessile drop, where alginate dispersed structures beneath the CHB/water interface are optically well resolved (Supplementary Fig. 10b,c and Supplementary Movie 2). Interestingly, the size of alginate droplets exhibited a characteristic transient evolution during the early stage (0–1 h). The mean droplet size initially increased within the first 10 min and subsequently decreased over the next several hours (Supplementary Fig. 10d). We attribute this behavior to partial coalescence among weakly crosslinked alginate droplets, producing larger and unstable droplets that rise to the interface and

merge into the external aqueous phase. Meanwhile, smaller alginate droplets remain in the CHB phase for longer, allowing sufficient contact with diffusing Ba$^{2+}$ ions to build surface charge and form stable hydrogel particles that eventually assemble into ordered colloidal structures.

### Discussion

The alginate–CHB platform developed here provides a controlled route to study interaction-driven ordering in a nonequilibrium colloidal environment. As Ba$^{2+}$ ions diffuse into the CHB phase and progressively crosslink the alginate droplets, surface charge builds up, strengthening the screened electrostatic repulsion that drives the transition from a disordered state to an ordered structure. By calibrating BD simulations to the experimentally measured lattice spring constant, we estimate the Debye screening length as $\kappa^{-1} \approx 2.5$–3 µm in the alginate colloid system. This estimate remains robust when particle-size dispersity is incorporated into the simulations, indicating that the spring constant primarily reflects the underlying interaction scale rather than size heterogeneity. The inferred screening length also falls within the range reported previously for colloidal systems in low-dielectric media[18,33–36].

Building on this interaction length scale, we use the dimensionless interaction parameter $\Gamma$ to summarize the time-dependent strengthening of electrostatic interactions during ordering. As surface charge builds up through ion diffusion and alginate hydrogelation, the resulting electrostatic repulsion strengthens, leading to an increase in $\Gamma$ with incubation time. $\Gamma$ reaches values of ≈117–149 at $t \approx 6$ h for $\kappa^{-1} \approx 2.5$–3 µm, coinciding with the experimentally observed onset of hexagonal ordering. At later times ($t \gtrsim 8$ h), $\Gamma$ exceeds ≈220, corresponding to a strongly interaction-dominated regime in which robust

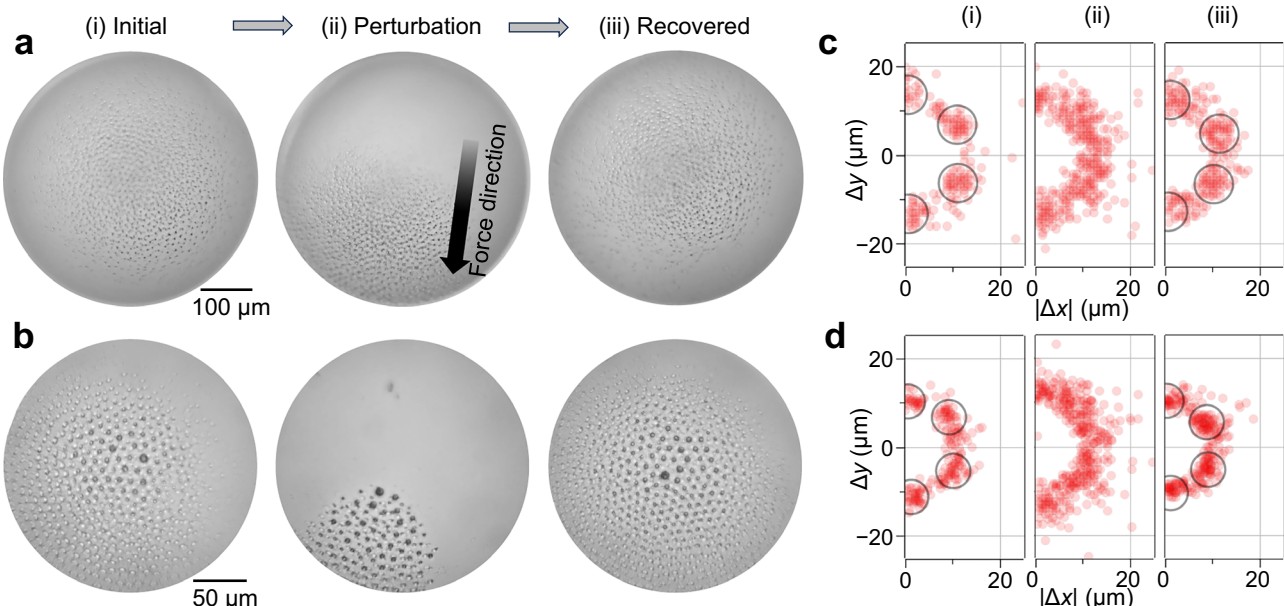

**Fig. 6 | Reversible response of magnetically functionalized alginate colloidal ordering to external perturbations. a, b** Bright-field images of the colloidal assembly at three stages: (i) initial ordered state, (ii) under perturbation, and (iii) after removal of the perturbation and recovery. **a** Magnetic field applied (arrow indicates force direction); **b** Gentle mechanical agitation. **c, d** Corresponding neighbor-displacement maps ($\Delta x$, $\Delta y$). Black circles represent regions with higher particle localization probability. Source data are provided as a Source Data file.

colloidal ordering is established. At each incubation time, particle configurations appear to relax on timescales much shorter than those associated with surface-charge buildup and screening evolution, allowing the system to reach quasi-steady structural states despite the continuous change in interaction strength. This separation of timescales justifies the use of equilibrium-based structural descriptors—such as the lattice spring constant and $\Gamma$—at each stage, even though the overall ordering process remains intrinsically nonequilibrium.

Beyond providing quantitative insights into interaction-driven ordering, the alginate–CHB platform offers practical advantages for studying soft matter systems under weakly ionic conditions. Stable ordered structures arise from straightforward and accessible steps—formation of alginate sub-droplets in CHB followed by incubation in contact with a $Ba^{2+}$-containing solution—without requiring complex particle synthesis. Although complete reproducibility is limited by the inherently nonequilibrium nature of the system, the platform remains a convenient and adaptable testbed for time-resolved studies under varied conditions, owing to its electrostatic tunability and compatibility with real-time optical imaging. Looking ahead, several directions could further deepen the mechanistic understanding of the observed ordering. Incorporating single-particle dynamical analyses, such as mean-squared displacements and cage escape times, would provide direct insight into relaxation pathways beyond the structural evolution emphasized here. Microscale temperature control could enable systematic exploration of thermal stability, while dilute-condition direct-force measurements between alginate hydrogels—such as optical tweezer–based interaction profiling[4,18]—could further refine interaction models and clarify melting pathways. In addition, although $Ba^{2+}$ ions clearly penetrate the CHB medium to crosslink alginate droplets, the microscopic mechanisms governing this ion transfer remain incompletely understood. Future studies quantifying interfacial partitioning, reaction–diffusion kinetics, and counterion exchange will be essential for determining how charge buildup and screening evolve in weakly dissociative media. Together, these efforts would provide a deeper mechanistic foundation for electrostatically mediated colloidal ordering under nonequilibrium conditions and broaden the framework for understanding charge-regulated assembly in low-dielectric environments.

## Methods

### Materials

Sodium alginate, iron oxide nanoparticles ($Fe_2O_3$, <50 nm), barium acetate ($Ba(C_2H_3O_2)_2$, $Ba(Ac)_2$, 99%), calcium chloride ($CaCl_2$, ≥93.0%), poly(vinyl alcohol) (PVA, $M_w$ = 13 – 23 kg mol$^{-1}$), chloroform ( ≥ 99.8%), toluene ( ≥ 99.5%), and trichloro(octadecyl)silane (OTS, ≥93.0%) were purchased from Sigma-Aldrich, USA. Cyclohexyl bromide (CHB, 98%) was obtained from Alfa Aesar, USA, and was filtered through aluminum oxide ($Al_2O_3$, pore size = 58 Å, Sigma-Aldrich, USA) to remove impurities before use. Potassium acetate ($CH_3COOK$, 95%) and sodium chloride (NaCl, 99.5%) were obtained from Duksan, Korea, and Samchun, Korea, respectively. Deionized water (DI water, resistivity ≥18.2 MΩ cm, Aquapuri5, Youngin, Korea) was always used to prepare the aqueous phase.

### Fabrication of microfluidic device

To generate monodisperse oil phase emulsion droplets in an aqueous continuous medium, we used a glass capillary co-flow device[19,20,40,41]. The device was assembled in a coaxial configuration on a glass slide (52 × 76 mm$^2$, Matsunami, Japan). A 50 mm circular glass capillary (inner diameter (I.D.) = 0.58 mm, outer diameter (O.D.) = 1.0 mm, World Precision Instruments, USA) was tapered using a micropipette puller (Model P-1000, Sutter Instrument, USA), then trimmed and polished using a microforge (MF-830, Narishige, Japan) and fine sandpaper to yield a nozzle orifice ≈120 μm in diameter. A 40 mm square capillary (inner diameter, ID = 1.05 mm, AIT Glass, USA) was fixed to the slide with optical adhesive (NOA81, Norland Products, USA), and the prepared inlet circular capillary was inserted concentrically into the square capillary. A 100 mm circular outlet capillary was inserted from the opposite end of the square capillary so that the tapered orifice of the inlet capillary was positioned slightly inside the outlet capillary. The inlet and outlet capillaries were then coaxially aligned within the square capillary and fixed in place on the slide using epoxy (5-min Epoxy, Devcon, USA). The opposite end of the outlet capillary was gently heated and slightly bent. A 20-gauge needle (KDS2012P, Weller, USA) was positioned in the gap between the square capillary and the inlet circular capillary; all remaining gaps were sealed with epoxy and

allowed to cure overnight. The inner circular capillary and the needle were connected to low-density polyethylene tubing (ID = 0.8636 mm, Scientific Commodities, Inc., USA) to serve as ports for introducing the inner and continuous phase solutions, respectively. Prior to droplet generation, the assembled device was flushed with DI water to remove particulates and residues from the channels.

## Preparation of alginate-nested CHB emulsions

Sodium alginate (1 wt.%) was dissolved in DI water and mixed with CHB at a 1:100 (v/v) ratio, then sonicated for 3 min to generate alginate aqueous droplets dispersed in CHB. The alginate-in-CHB dispersion was loaded into a 10 mL gas-tight syringe (SGE, Trajan Scientific and Medical, Australia), mounted vertically on a syringe pump (Pump 11 Elite, Harvard Apparatus, USA), and left for ≈15 min to allow the less dense alginate droplets to float upward within the syringe. The dispersion was then supplied as the inner (dispersed) phase to the co-flow microfluidic device, while the outer (continuous) phase—DI water containing 2 wt.% PVA for emulsion stabilization—was delivered from a 50 mL gas-tight syringe on a separate pump. Flow rates were set to 50 μL min$^{-1}$ (inner) and 500 μL min$^{-1}$ (outer), yielding CHB droplets in water that encapsulate numerous alginate sub-droplets (Alg@CHB). CHB droplet generation was monitored on an inverted optical microscope (Eclipse TS100, Nikon, Japan) equipped with a high-speed camera (Miro eX2, Phantom, USA). After confirming stable production for ≈5–7 min, the CHB droplets were collected into a cylindrical chamber containing the desired ion solution. The chamber was constructed by bonding a glass ring to a circular coverslip (Marienfeld, No. 1.5H, Germany) using a UV-curable adhesive (Optical Adhesive 81, Norland, USA). The assembled chamber was thoroughly cleaned with ethanol and DI water, followed by oxygen plasma treatment for 30 s using a plasma cleaner (PDC-32G-2, Harrick Plasma, USA). Because the collected CHB droplets are dispersed in the aqueous phase, they do not come into contact with the UV adhesive, thereby eliminating the possibility of impurity generation in CHB through adhesive dissolution[18]. For experiments requiring magnetic responsiveness, iron-oxide nanoparticles were added to the 1 wt.% alginate solution to a final concentration of 5 mg mL$^{-1}$ prior to mixing with CHB, and the same procedure was followed.

## Image analyses of alginate colloidal ordering

Quantitatively analysis was performed in ImageJ/FIJI (1.54r)[42]. From $z$-stacked optical microscope images (IX83/FV3000, Olympus, Japan) acquired with a charge-coupled device (CCD) camera (DP80, Olympus, Japan), particle centroids were identified and their $z$ positions assigned from the motorized stage displacement recorded during sequential acquisition. Using the centroids in a given analysis plane (layer), Delaunay triangulation provided interparticle distances and nearest-neighbor lists; from these we computed the mean center-to-center spacing $\langle r_{cc} \rangle$ and generated neighbor-displacement maps $\Delta \mathbf{r} = (\Delta x, \Delta y)$ used to visualize the nearest-neighbor shell and angular clustering. Voronoi tessellation was then used to access local geometric order by partitioning space into polygons associated with individual particles; the coordination-number distribution (fraction of 5, 6, 7-fold cells and its spread) served as a measure of hexagonal order. To evaluate global structural order, particle coordinates (or binary particle masks) were transformed by fast Fourier transform (FFT); sixfold spot patterns in the FFT indicated strong periodicity, whereas ring-like or smeared features reflected disorder or defect-rich states.

For 3D reconstruction, sequential image slices were stacked into a 3D TIFF in ImageJ/FIJI[42]. Machine-learning segmentation (Trainable Weka Segmentation) generated probability maps that were manually curated to remove artifacts or non-particle regions[43]. A Gaussian blur was applied to reduce noise while preserving structural details, and spherical aberration correction was performed assuming approximately spherical particle geometry. Particle centroids were then located in 3D, and maximum-inscribed-sphere fits were used to estimate particle diameters and separations. The resulting 3D coordinates enabled depth-resolved analyses.

## Zeta potential of alginate hydrogel particles in CHB

Measuring the zeta potential $\zeta$ of alginate hydrogel particles directly inside CHB-in-water emulsions containing barium ions is nontrivial. To isolate the effect of $Ba^{2+}$ while enabling reliable electrophoretic measurements in the CHB phase, we adopted a planar oil–water geometry. A 1 wt.% aqueous sodium alginate stock and a 200 mM $Ba(Ac)_2$ stock were prepared separately. Alginate sub-droplets in CHB were generated by mixing the alginate solution with CHB at 1:100 (v/v) ratio followed by bath sonication for 3 min to obtain a uniform dispersion. For time-series measurements, 10 mL of the alginate–CHB dispersion was aliquoted into eight identical glass vials. To establish a controlled oil–water interface, 5 mL of the $Ba(Ac)_2$ solution was gently layered onto each CHB aliquot, and the incubation time $t$ was counted from this moment. Vials were left undisturbed except for gentle manual agitation at the predefined sampling times (15 min, 30 min, 1 h, 2 h, 4 h, 6 h, 12 h, and 24 h). After each incubation period, the denser CHB phase was carefully withdrawn from the bottom of the vial, avoiding aqueous carry-over, and transferred to a Zetasizer (ZEN3600, Malvern Instruments, UK). Electrophoretic mobility was converted to $\zeta$ using the Hückel approximation, which has been used in CHB-based colloidal systems due to the low ionic strength and the resulting long-ranged electrostatic environment. Prior studies on PMMA colloids dispersed in CHB–decalin or CHB–n-decane mixtures—such as Royall et al.[33] and our previous measurements[18]—have demonstrated that the Hückel formulation remains robust in these weakly screened nonpolar media.

## Brownian dynamics

We estimated the effective lattice spring constant of a 2D colloidal array by overdamped BD using the discrete (Euler-Maruyama) Langevin update[44,45]. Spherical particles of radius $a$ interacted via a screened Coulomb (Yukawa) potential, $U(r) = \frac{U_0 e^{-\kappa r}}{\kappa r}$ with $U_0 = 4\pi \varepsilon_0 \varepsilon \zeta^2 R^2 \kappa e^{2\kappa R}$ [34–36,46], where $\varepsilon_0$ is the vacuum permittivity, $\varepsilon$ is the dielectric constant of CHB, $\zeta$ is the zeta potential, and $\kappa^{-1}$ is the Debye screening length. The Debye length was parameterized as $\kappa^{-1} = \sqrt{\frac{\varepsilon_0 \varepsilon k_B T}{e^2 N_A I}}$, where e is the elementary charge, $N_A$ is the Avogadro's number, $k_B T$ is the thermal energy, and $I$ is the ionic strength. For barium acetate, which dissociates into one $Ba^{2+}$ and two $C_2H_3O_2^-$, the ionic strength is calculated as $I = 3000 C_{Ba(Ac)_2}$ in mol m$^{-3}$. The pair force on particle $i$ due to $j$ was $\mathbf{F}_{ij} = -\nabla_i U(r_{ij}) = U_0 \frac{e^{-\kappa r_{ij}}}{\kappa r_{ij}^2} \left( 1 + \kappa r_{ij} \right) \hat{\mathbf{r}}_{ij}$, where $\mathbf{r}_{ij} \equiv \mathbf{r}_i - \mathbf{r}_j$ is the center-center vector, $r_{ij} \equiv |\mathbf{r}_{ij}|$ its magnitude, and $\hat{\mathbf{r}}_{ij} \equiv \mathbf{r}_{ij}/r_{ij}$ the unit vector from $j$ to $i$. Particle positions obeyed the discrete overdamped Langevin scheme, $\mathbf{r}_i(t + \Delta t) = \mathbf{r}_i(t) + \frac{\mathbf{F}_i \Delta t}{\gamma} + \sqrt{\frac{2 k_B T \Delta t}{\gamma}} \boldsymbol{\xi}_{i,t}$, with Stokes drag $\gamma = 6\pi R \eta$ (CHB viscosity $\eta$), and independent Gaussian noises $\boldsymbol{\xi}_{i,t} \sim \mathcal{N}(\mathbf{0}, \mathbf{I}_2)$, where $\mathbf{I}_2$ is the 2 × 2 identity. We used $\Delta t = 10^{-3}$ s, periodic boundaries with the minimum-image convention, and $N_p$ particles initialized on a square lattice (box length $L$) with a small random jitter. After an equilibration of 200 s, coordinates were saved every $30^{-1}$ s over an observation window $t_w$. For Fig. 5, we set $N_p = 25$, $L = 62.9$ μm, $R = 1.42$ μm, and $t_w = 65$ s to match the experimental acquisition used for $k_s^{exp}$. To extract $k_s^{BD}$, trajectories were unwrapped and analyzed in 65 s windows. In each frame we removed global drift (center-of-mass subtraction) and global rotation (Kabsch alignment) to isolate cage-relative fluctuations[29–31,47,48]. For particle $i$, the cage center $\langle \mathbf{r}_i \rangle$ was the time average within the window, and the cage-relative displacement was $\mathbf{u}_i(t) = \mathbf{r}_i(t) - \langle \mathbf{r}_i \rangle$. Assuming small-amplitude vibrations in an approximately harmonic cage, equipartition in 2D yields $k_s = \frac{2 k_B T}{\langle u^2 \rangle}$, $\langle u^2 \rangle = \langle u_x^2 + u_y^2 \rangle$, with the average over particles and time

samples in the window. We report $k_s^{BD}$ as mean ± standard deviation over ten independent runs. All simulations and analyses were implemented in MATLAB (R2025a).

## Experimental determination of the lattice spring constant

Bright-field videos were acquired at 30 frames per second (fps) using a CCD camera (DP80, Olympus, Japan) mounted on an optical microscope (IX83/FV3000, Olympus, Japan). Particle positions were obtained in ImageJ by defining a region of interest (ROI) and applying threshold-based segmentation to extract centroid coordinates $(x,y)$ in each frame. Subsequent processing and $k_s^{exp}$ estimation were identical to the BD analysis.

## Validation of the 2D spring-constant estimator

The spring-constant estimator was validated with a single-particle harmonic-trap test. We simulated a single overdamped Brownian particle in a known 2D harmonic trap by overdamped BD using the discrete Langevin update, and processed the trajectory with the same analysis method used for BD/experiment. The trap potential was $V(\mathbf{r}) = \frac{1}{2} k_s^{input} |\mathbf{r}|^2$. Positions were advanced as $\mathbf{r}(t + \Delta t) = \mathbf{r}(t) - \frac{k_s^{input}\mathbf{r}(t)\Delta t}{\gamma} + \sqrt{\frac{2k_B T \Delta t}{\gamma}}\boldsymbol{\xi}_t$, with Stokes drag $\gamma = 6\pi R\eta$, and independent Gaussian noise $\boldsymbol{\xi}_t \sim \mathcal{N}(\mathbf{0}, \mathbf{I}_2)$. Within each analysis window, we formed cage-relative displacements $\mathbf{u}(t) = \mathbf{r}(t) - \langle \mathbf{r} \rangle$ (time average over the window) and estimated $k_s^{BD} = \frac{2k_B T}{\langle u^2 \rangle}$, $\langle u^2 \rangle = \langle u_x^2 + u_y^2 \rangle$ with averages over time samples in the window.

## Particle-size dispersity in BD simulations

To examine how particle-size dispersity affects the microstructure and the effective lattice spring constant ($k_s$) of the colloidal assembly structure, we performed BD simulations with $N = 100$ particles interacting via a screened-Coulomb potential. The box size was chosen such that the surface coverage matched that used in Fig. 5a-c. Experimentally, the top-layer alginate hydrogel particles in Fig. 5a exhibit a mean radius $\langle R \rangle_{exp} \approx 1.42$ μm and a standard deviation $\sigma_{R, exp} \approx 0.19$ μm, corresponding to a coefficient of variation, $CV_{exp} = \frac{\sigma_{R, exp}}{\langle R \rangle_{exp}} \approx 0.13$. To implement controlled dispersity while ensuring strictly positive radii, we sampled particle sizes from a Gamma distribution, $R \sim \Gamma(k, \theta)$[37,38], where the shape parameter $k$ and scale parameter $\theta$ satisfy a mean $\mu = \langle R \rangle = k\theta$ and a variance $\sigma^2 = k\theta^2$. The coefficient of variation is $CV = \frac{\sigma}{\mu} = \frac{1}{\sqrt{k}}$, which gives $k = \frac{1}{CV^2}$ and $\theta = CV^2\mu$. In simulations, we set the target mean radius to $R_0 = 1.42$ μm and varied the imposed CV. We then sampled $R_i \sim \Gamma(k, \theta)$, $i = 1, \ldots, N$.

Particles were initialized on a slightly perturbed square lattice in a periodic box. Each particle experienced overdamped Brownian dynamics with drag coefficient $\gamma_i = 6\pi\eta R_i$ and diffusion coefficient $D_i = \frac{k_B T}{\gamma_i}$. The pairwise Yukawa interaction between particles $i$ and $j$ at separation $r_{ij}$ had effective amplitude, $U_{0,ij} = 4\pi\varepsilon_0\varepsilon\zeta^2\kappa R_i R_j e^{\kappa(R_i + R_j)}$, and force $F_{ij}(r_{ij}) = U_{0,ij} \frac{e^{-\kappa r_{ij}}}{\kappa r_{ij}^2}(1 + \kappa r_{ij})$. After an equilibration period, particle trajectories were recorded at fixed time intervals.

## Colloidal ordering in a flattened sessile drop

Sodium alginate (1 wt.%) was dissolved in DI water and mixed with CHB at a volume ratio of 0.1:100. The mixture was sonicated for 3 min to generate alginate aqueous droplets dispersed in the CHB phase. Separately, an aqueous phase containing Ba(Ac)$_2$ and PVA was prepared by mixing 40 mM Ba(Ac)$_2$ and 2 wt.% PVA at a 2:1 volume ratio. A cylindrical chamber was constructed by bonding a glass ring to a circular coverslip (Marienfeld, No. 1.5H, Germany) using a UV-curable adhesive (Optical Adhesive 81, Norland, USA). The chamber was subsequently modified to enable the formation of a flattened CHB sessile drop within the aqueous phase. The bottom glass surface was rendered hydrophobic by applying 0.5 wt.% OTS in toluene, followed by drying in a heated oven for several minutes. After surface treatment, the chamber was filled with 2 mL of the Ba(Ac)$_2$/PVA aqueous solution. A few microliters of the alginate–CHB dispersion was then introduced onto the hydrophobic substrate, where the CHB phase spread into a flattened lens-shaped droplet within the aqueous phase (Supplementary Fig. 10a). The three-phase contact angle of this CHB sessile drop could not be measured using a tensiometer (Attension Theta Auto 1B, Biolin Scientific, Sweden), indicating that the CHB/water interface remained effectively flat while still providing geometric confinement for the formation of ordered alginate hydrogel particles within the CHB phase. Importantly, the CHB sessile drop remained localized on the hydrophobic central region and did not reach the adhesive-sealed boundary, ensuring that it never contacted the UV-curable adhesive and eliminating the possibility of impurity generation through adhesive dissolution[18].

## Reporting summary

Further information on research design is available in the Nature Portfolio Reporting Summary linked to this article.

## Data availability

The data that support the findings of this study are available from the corresponding authors upon request. Unprocessed raw data are provided as Supplementary Data 1. Source data are provided with this paper.

## Code availability

The codes used in this study are available from the corresponding authors upon request. The custom BD simulation codes (m-files) used for Fig. 5b, c and Supplementary Fig. 9 are provided as Supplementary Data 2.

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

## Acknowledgements

This study was supported by the National Research Foundation (NRF) of Korea (RS-2025-00516792 to B.J.P., RS-2024-00406741 to B.J.P., and RS-2025–16063688 to H.H). The research was conducted with the support of the equipment and facilities provided by the ACE Center at Kyung Hee University.

## Author contributions

B.J.P. conceived and supervised the project. I.H.J. C.C.R. and H.S.L. performed the microfluidic preparation and measured the behaviors of colloidal ordering. C.C.R. measured the zeta-potential of alginate hydrogel particles. I.H.J. and H.A. performed the image analysis. B.J.P. performed the BD simulations. H.H., H.A., and B.J.P. wrote the first draft. All authors contributed to the interpretation of experimental data and read, edited, commented on this manuscript.

## Competing interests

The authors declare no competing interests.
