## [Transparent Peer Review file · Nature Communications]

Nonequilibrium ordering dynamics of confined soft alginate hydrogel colloids driven by time-evolving electrostatic interactions

Corresponding Author: Professor Bum Jun Park

Version 0:

Reviewer comments:

Reviewer #1

(Remarks to the Author)

Authors the manuscript use experiments and simulations to study the formation as well as dynamics of nonequilibrium Wigner-type crystal using colloids as model system. Experiments and analysis are sound and can be potentially published. However, the manuscript requires major revision. One of the concerns that I have is that the system size considered is very small and I wonder what would happen if the system size is larger - for example - planar interfaces. There are also other issues related to details and presentation that needs to be considered/addressed:

(1) The Hückel approximation used for zeta-potential calculation is only valid at low potentials and does not apply, especially at high incubation time (see Supplementary Fig. 2)

(2) I understand that authors have developed an alternate strategy for zeta potential measurement. However, the method described is quite complex and yet does not represent the actual experimental conditions corresponding to observation of Wigner-Type Crystals.

(3) I presume alginate particles are soft and also deformable, what is the rational for modeling interparticle interaction with Yukawa (screened-Coulomb) potential, which is typically used for hard colloids.

(4) Authors should justify the need for using nested emulsions. This could have also been done by considering planar interfaces.

(5) Since the alginate particles are generated via microfluidic technique coupled with cross-linking, monodisperse particles are expected. However, authors discuss some of the results based on size segregation and particle size effect. This needs to be clarified.

(6) As per Figure 5(d), zeta potential is positive whereas in Figure 2 in the supporting file, it is negative. This is contradicting reporting.

(Remarks on code availability)

Reviewer #2

(Remarks to the Author)

This study presents a clever and experimentally accessible colloidal analogue of Wigner crystallization of electrons but then using soft alginate particles dispersed in cyclohexyl bromide (CHB). By tuning electrostatic screening and monitoring lattice formation, the system reproduces key scaling features of electron Wigner crystals, notably the critical Wigner parameter ($\Gamma \approx 120-150$) at the onset of ordering. The consistent correspondence between measured and simulated lattice properties

underscores the robustness of screened electrostatic interactions even in this nonequilibrium environment.

Overall, the work provides a compelling and well-constructed analogy between colloidal and electronic Wigner crystallization. Conceptually, it succeeds in showing how a universal organizing principle—the dominance of electrostatic repulsion over thermal motion—governs lattice formation across very different physical systems. Quantitatively, the study demonstrates impressive internal consistency: the experimentally determined Wigner parameter and screening length coincide with both simulations and prior literature, lending credibility to the physical interpretation. The experimental platform itself is another strength—simple, tunable, and compatible with real-time imaging—offering a practical means to study charge-driven self-assembly. Nonetheless, several limitations remain. The simulations omit polydispersity and interaction heterogeneity, which likely overstate lattice stability; the physical origin and temporal stability of screening in CHB remain uncertain (see further); and the nonequilibrium nature of the system introduces reproducibility challenges. Looking forward, the framework holds significant promise, especially if complemented by direct measurements of ionic dissociation and particle interactions. Such refinements could transform this elegant soft-matter analogue into a quantitatively reliable model for exploring Wigner-type ordering and melting phenomena.

One key concern regards the solvent used. CHB is only weakly dissociative, and its ionic content—and hence the effective screening length ($\kappa^{-1} \approx 3 \mu\text{m}$)—depends sensitively on trace impurities, counterion exchange, and residual water or surface active component contamination and possibly time. The chemical stability and reproducibility of the electrostatic environment with CHB has been discussed in literature for colloidal crystals of PMMA, and the properties of CHB which may evolve over hours as ions partition between the alginate network and the solvent and back to batch differences between CHB solvents. While the framework offers a practical route to visualize charge-driven ordering, a clearer understanding of ion dissociation and mobility in CHB should be discussed.

(Remarks on code availability)

Reviewer #3

(Remarks to the Author)

The manuscript by Jung et al. reports the formation of ordered hexagonal patterns in charged alginate hydrogel microparticles confined within CHB droplets, which the authors interpret as a room-temperature “model Wigner-type crystal.” The system shows interesting colloidal self-assembly: Ba²⁺-mediated crosslinking

increases repulsion over time, buoyancy-driven compaction yields quasi-2D layering, and hexagonal order is quantified via Voronoi/FFT metrics. They also show reversible disorder–order after magnetic/mechanical perturbations.

However, I have major concerns about the central premise and interpretation, leading me to recommend REJECT. The work is framed as a “macroscopic, real-time analogue” of electronic Wigner crystallization and draws explicit parallels to electron systems. This analogy is not physically justified: it conflates interaction-driven localization of mobile carriers with equilibrium self-assembly of intrinsically localized charged objects. These are not issues that can be addressed through revision, they reflect a fundamental mischaracterization of physics that would mislead readers about the nature of Wigner crystallization.

A Wigner crystal, by definition, arises when free, mobile charge carriers (electrons) become spatially ordered because Coulomb repulsion energy overwhelms their kinetic energy—the very kinetic energy that would otherwise allow them to remain itinerant. Classic realizations—electrons trapped on liquid helium surfaces (Grimes & Adams, Phys. Rev. Lett. 42, 795, 1979) or magnetically induced Wigner solids in ultra-clean two-dimensional electron gases (Andrei et al., Phys. Rev. Lett. 60, 2765, 1988)—involve no underlying lattice and no external forces. Ordering emerges purely from the competition between interaction and kinetic energies of the carriers themselves.

The present system bears no resemblance to this physics. The charges are intrinsically localized in crosslinked polyelectrolyte droplets from the outset. Just like apples in a box. There are no mobile carriers, no electronic band structure, no hopping transport, and no itinerant kinetic energy to overcome. The system forms a nonequilibrium colloidal crystal under time-dependent charging (progressive Ba²⁺ crosslinking) and external forces (buoyancy-driven stratification), not through a thermodynamic phase transition driven by interaction-kinetic energy competition. What we observe is simply a colloidal Yukawa lattice—well-studied soft-matter self-assembly—where pre-localized objects arrange under repulsive interactions and gravity. Any insulator has localized electrons, this does not make every ordered insulating structure a Wigner crystal.

The authors' definition of the Wigner parameter Γ also exemplifies the conceptual confusion. They define Γ as the ratio of Yukawa interaction energy to

Brownian thermal energy of colloidal particles. In electronic Wigner physics, the relevant energy scale is electronic kinetic energy—Fermi energy for quantum systems, or kinetic energy of mobile point charges in classical plasmas. Equating Brownian fluctuations of permanently localized droplets with electronic kinetic energy is invalid. The authors note their measured $\Gamma \sim 117\text{--}149$ matches electronic Wigner crystal thresholds ($\Gamma \sim 95\text{--}150$) and present this as evidence, but numerical coincidence means nothing when the underlying degrees of freedom, conservation laws, and dynamical processes differ categorically.

The authors claim their platform enables observation of "electronic behavior at macroscopic scale," but Wigner crystallization physics does not scale trivially with particle size. Quantum mechanics itself cannot be reproduced by enlarging particles and substituting thermal energy for electronic kinetic energy. The authors provide no rigorous dimensional analysis or dynamical mapping to justify that their system preserves the relevant physics.

While the experimental observations are competently executed, the central claim—that this system realizes or analogizes electronic Wigner crystallization—is not physically justified. The distinction between interaction-driven localization of mobile carriers and structural self-assembly of fixed charged objects is fundamental physics, not interpretation. The authors should consider re-preparing this work reframed as a study of colloidal crystal formation under screened electrostatic interactions, time-dependent charging, and geometric confinement, without invoking the Wigner crystal analogy their system cannot support.

(Remarks on code availability)

Reviewer #4

(Remarks to the Author)

(Remarks on code availability)

Reviewer #5

(Remarks to the Author)

(Remarks on code availability)

Version 1:

Reviewer comments:

Reviewer #1

(Remarks to the Author)

There appears to be several aspects overlooked in the revised version of the manuscript. Therefore, I am not in favor of publication of the manuscript in current form and suggest major revision.

(1) The mechanism of formation of nested droplets is unclear. Are the droplets of sodium alginate already present in CHB prior to it coming in contact with aqueous PVA solution? What drives the nested emulsion formation? kinetics or thermodynamics?

(2) I looked through the citation pertaining to explanation for change in zeta potential presented on Page 6, *Biomacromolecules* 2006, 7, 5, 1471–1480. This article does not discuss anything about zeta or charge measurements. Moreover, the cross-linking process is supposed to reduce charge and hence zeta potential is supposed to decrease. Therefore, the explanation provided does not seem correct, although I believe in the data presented.

(3) The buoyancy-driven stratification must be substantiated with density measurements or rising velocity calculations

(4) Figure 4 (a) and (b) - correspondence between particles in (b) identified from analysis of particles in (a) appears poor. Some of the droplets in the images are not in focus, therefore, the image analysis and further calculations need to be checked.

(5) Debye length calculated (Page 13 and Figure 5), which are or the order of few microns, seem unusually high and do not make physical sense. Please consider including reason for this.

(6) In Figure 10 (c) in Supplementary, although enlarged and contrast-enhanced images are shown, the emergence of triangular order is not there, contrary to what is claimed. Therefore, the discussion presented in the section "Alginate colloidal ordering in a flattened sessile drop geometry", lacks clarity and brings a question about generality of the observations made in the droplets.

(Remarks on code availability)

Reviewer #2

(Remarks to the Author)

The manuscript presents an experimentally elegant colloidal system that reproduces several key phenomenological features of Wigner crystallization using soft alginate particles in a low-dielectric solvent. The identification of a critical interaction parameter ($\Gamma \approx 120-150$) at the onset of ordering, supported by calibrated Brownian dynamics simulations, is a clear strength and lends quantitative credibility to the analogy.

The newly added analysis of particle-size polydispersity appropriately demonstrates that structural order is sensitive to disorder, while the inferred interaction length scale remains comparatively robust. Nevertheless, the system remains intrinsically nonequilibrium, and the microscopic origin, temporal stability, and reproducibility of electrostatic screening in CHB remain somewhat qualitative. While the separation of timescales provides a reasonable justification for using equilibrium-based structural descriptors, this assumption should be borne in mind when interpreting the results. Overall, the study provides a well-constructed and physically transparent soft-matter analogue of charge-driven crystallization, while also highlighting important directions for future mechanistic refinement (CHB related). The latter could be asked for now or left for future work

(Remarks on code availability)

Reviewer #4

(Remarks to the Author)

The revised manuscript by Jung et al. presents a simple and elegant experimental platform to study colloidal ordering driven by electrostatic interactions. The manuscript has improved substantially in clarity and framing, including the title, abstract, and overall narrative. I particularly appreciate the addition of the sessile-drop experiments, which significantly strengthen the robustness and generality of the results.

I recommend publication in Nature Communications, subject to the following minor comments:

1. The terms triangular and hexagonal ordering are used interchangeably throughout the manuscript. Given that the experiments clearly show dominant sixfold symmetry, I suggest consistently using hexagonal ordering. While triangular motifs underlie a hexagonal lattice, the observed lattice structure is hexagonal, and consistent terminology would improve clarity.

2. The statement in line 76 ("Although particle ... nonequilibrium") would benefit from further clarification. It is not immediately clear how structural relaxation could proceed faster than the underlying charge buildup that governs the evolution of interaction strength. This point appears closely related to the statement in line 130 ("the magnitude of the zeta potential ... ordering kinetics," Fig. 1d), which suggests a coupling between charge buildup and ordering dynamics. A brief clarification reconciling these statements would help improve the physical interpretation.

(Remarks on code availability)

REVIEWER COMMENTS

Reviewer #1 (Remarks to the Author):

Authors the manuscript use experiments and simulations to study the formation as well as dynamics of nonequilibrium Wigner-type crystal using colloids as model system. Experiments and analysis are sound and can be potentially published.

We thank the reviewer for the positive assessment of our experimental design, analysis, and the overall potential of our study. We appreciate the constructive feedback provided, and we have carefully addressed each comment in detail below.

However, the manuscript requires major revision. One of the concerns that I have is that the system size considered is very small and I wonder what would happen if the system size is larger - for example - planar interfaces.

We thank the reviewer for this insightful comment. To examine whether alginate colloidal ordering persists under conditions of lower interfacial curvature, where the system size is effectively much larger, we conducted additional experiments using a flattened CHB sessile drop covered by an aqueous Ba^{2+} solution (Supplementary Fig. 10). In this geometry, triangular ordering still developed after ~8 hr, consistent with the behavior observed in the confined CHB droplet system. Note that some degree of geometric confinement is required for the ordering process; therefore, we employed such a sessile drop configuration while rendering the substrate hydrophobic to ensure good CHB wetting and formation of an interface with a sufficiently large radius of curvature. These results indicate that the observed ordering is not limited by the small system size of the original droplet geometry and can robustly emerge even under much larger interfacial dimensions. The description of the sessile drop experiment has been added to the main text, the Methods section, and the Supplementary Information for clarity.

Newly added section (p17):

Alginate colloidal ordering in a flattened sessile drop geometry

In addition to the confined CHB droplets discussed above, we verified that alginate colloidal ordering also emerges in a flattened CHB sessile drop geometry with a much larger interfacial radius of curvature (Supplementary Fig. 10a). Despite the altered geometry, the system exhibited a similar time scale for ordering: well-defined hexagonal domains appeared in the CHB phase approximately 8 hr after exposure to Ba^{2+} , consistent with the droplet-based experiments. This demonstrates that the ordering mechanism is robust and not restricted to the confined droplet environment, and that the sessile drop approach effectively illustrates the generality of the phenomenon. These observations also support the use of vial-based ζ -potential measurements to describe the electrostatic charging behavior of alginate hydrogel particles in both geometries.

To clearly visualize the ordering process, we monitored the edge region of the CHB sessile drop, where alginate dispersed structures beneath the CHB/water interface are optically well resolved (Supplementary Fig. 10b,c). Interestingly, the size of alginate droplets exhibited a characteristic transient evolution during the early stage (0–1 hr). The mean droplet size initially increased within the first 10 min and subsequently decreased over the next several hours (Supplementary Fig. 10d). We attribute this behavior to partial coalescence among

[Type here]

weakly crosslinked alginate droplets, producing larger and unstable droplets that rise to the interface and merge into the external aqueous phase. Meanwhile, smaller alginate droplets remain in the CHB phase for longer, allowing sufficient contact with diffusing Ba^{2+} ions to build surface charge and form stable hydrogel particles that eventually assemble into ordered colloidal structures.

Newly added section (p23):

Colloidal ordering in a flattened sessile drop

Sodium alginate (1 wt%) was dissolved in DI water and mixed with CHB at a volume ratio of 0.1:100. The mixture was sonicated for 3 min to generate alginate aqueous droplets dispersed in the CHB phase. Separately, an aqueous phase containing $\text{Ba}(\text{Ac})_2$ and PVA was prepared by mixing 40 mM $\text{Ba}(\text{Ac})_2$ and 2 wt% PVA at a 2:1 volume ratio. A cylindrical chamber, prepared as described above, was further modified to enable the formation of a flattened CHB sessile drop within the aqueous phase. The bottom glass surface was rendered hydrophobic by applying 0.5 wt% OTS in toluene, followed by drying the chamber in a heated oven for several minutes. After surface treatment, the chamber was filled with 2 mL of the $\text{Ba}(\text{Ac})_2$ /PVA aqueous solution. A few microliters of the alginate–CHB dispersion was then introduced onto the hydrophobic substrate, where the CHB phase spread into a flattened lens-shaped droplet within the aqueous phase (Supplementary Fig. 10a). The three-phase contact angle of this CHB sessile drop could not be measured using a tensiometer (Attension Theta Auto 1B, Biolin Scientific, Sweden), indicating that the CHB/water interface remained effectively flat while still providing geometric confinement for the formation of ordered alginate hydrogel particles within the CHB phase. Importantly, the CHB sessile drop remained localized on the hydrophobic central region and did not reach the adhesive-sealed boundary, ensuring that it never contacted the UV-curable adhesive and eliminating the possibility of impurity generation through adhesive dissolution.¹⁸

Newly added Supplementary figure (pS12 in SI):

[Type here]

Supplementary Fig. 10 Alginate colloidal ordering in a flattened CHB sessile drop geometry. **a** Schematic of the sessile drop setup, where alginate droplets in CHB are confined beneath an aqueous Ba^{2+} /PVA layer. **b** Bright-field images showing time-resolved ordering of alginate droplets at the edge region of the sessile drop. **c** Enlarged and contrast-enhanced views of the boxed regions in panel **b**, with FFT insets highlighting the emergence of triangular order. **d** Temporal evolution of alginate droplet size, with an inset showing magnified view of the early-time regime (0–1 hr).

There are also other issues related to details and presentation that needs to be considered/addressed:

(1) The Hückel approximation used for zeta-potential calculation is only valid at low potentials and does not apply, especially at high incubation time (see Supplementary Fig. 2)

We thank the reviewer for pointing out the limitations of the Hückel approximation at elevated surface potentials. We agree that the small-potential assumption of the Hückel model is not strictly satisfied when $\zeta \approx 50-60$ mV. However, in low-ionic-strength media such as CHB, the Debye length is comparable to or larger than the particle radius, meaning that electrostatic

[Type here]

screening is extremely weak and the double layer extends far into the solvent. Under these conditions, electrophoretic mobility depends only weakly on the specific electrokinetic model used, and the Hückel formulation has been adopted as a practical and robust method for converting mobility to ζ in CHB-based colloidal systems. For example, Royall et al. [J. Chem. Phys., 124, 244706 (2006)] used the Hückel equation to determine ζ for PMMA particles dispersed in CHB–decalin, noting that the long-ranged electrostatic environment in such low-ionic-strength media justifies use of the Hückel limit. Likewise, in our earlier work on PMMA particles in CHB/n-decane mixtures [Soft Matter, 15, 8051 (2019)], we employed the same Zetasizer-based method and obtained $\zeta \approx 55$ mV. Importantly, this measured ζ was also consistent with values inferred independently from fitting directly measured PMMA interparticle forces to the screened-Coulomb (Yukawa) potential. We have clarified these points in the Supplementary Information.

Before: Electrophoretic mobility was converted to ζ using the Hückel approximation. For each time point, at least seven independent measurements were acquired; means and standard deviations are reported (Supplementary Fig. 2). It is important to note that ion exchange and diffusion kinetics at the planar oil–water interface may not identically reproduce those in the nested-emulsion geometry used elsewhere in this study. Nevertheless, the vial protocol captures the time-dependent increase in $|\zeta|$, reflecting progressive Ba^{2+} -mediated crosslinking and surface-charge buildup on alginate hydrogel particles, and thus provides a consistent qualitative proxy for the strengthening electrostatic repulsion observed during crystallization.

After (pS3 in SI): Electrophoretic mobility was converted to ζ using the Hückel approximation, which has been used in CHB-based colloidal systems due to the extremely low ionic strength and the resulting long-ranged electrostatic environment. Prior studies on PMMA colloids dispersed in CHB–decalin or CHB–n-decane mixtures—such as Royall et al.¹ and our previous measurements²—have demonstrated that the Hückel formulation remains robust in these weakly screened nonpolar media. For each time point, at least seven independent measurements were acquired; means and standard deviations are reported (Supplementary Fig. 2). While ion exchange and diffusion kinetics at the planar oil–water interface may not identically reproduce those in the nested-emulsion geometry, the vial protocol reliably captures the time-dependent increase in $|\zeta|$, reflecting progressive Ba^{2+} -mediated crosslinking and surface-charge buildup on alginate hydrogel particles. Accordingly, these measurements provide a consistent qualitative proxy for strengthening electrostatic repulsion that underlies alginate colloidal ordering in our system.

(2) I understand that authors have developed an alternate strategy for zeta potential measurement. However, the method described is quite complex and yet does not represent the actual experimental conditions corresponding to observation of Wigner-Type Crystals.

We appreciate the reviewer’s concern regarding the difference between the ζ -potential measurement geometry and the actual environment in which alginate colloidal ordered structure is formed. As mentioned, we performed the additional experiments in the sessile drop

[Type here]

configuration, where the CHB phase forms a flattened interface with a much larger interfacial radius of curvature, approaching quasi-planar conditions (Supplementary Fig. 10). This geometry more closely resembles the ζ -measurement configuration while still allowing Ba^{2+} -mediated crosslinking and particle ordering to occur. Notably, we observed that triangular ordering emerged in the CHB sessile drop system with the similar characteristic timescale (~ 8 hr) as in the nested droplets (Fig. 2). This demonstrates that the time-dependent increase in $|\zeta|$ obtained from the vial-based measurements is consistent with the interparticle repulsion required for ordering in both geometries. These observations confirm that the ζ -potential measurements, although obtained in a simplified configuration, capture the essential charging dynamics relevant to the formation of alginate colloidal ordered structures.

Before: The magnitude of zeta potential ζ increased with incubation time, indicating progressive surface-charge buildup from Ba^{2+} -carboxylate interactions and continued crosslinking, consistent with the observed crystallization kinetics (Fig. 1d).

After (p6): The magnitude of zeta potential ζ increased with incubation time, indicating progressive surface-charge buildup from Ba^{2+} -carboxylate interactions and continued crosslinking,²⁸ consistent with the observed ordering kinetics (Fig. 1d). Although the vial geometry does not replicate the confined droplet environment exactly, it captures the same Ba^{2+} -mediated charging mechanism in the CHB phase and thus provides a reasonable effective measure of the evolving interaction strength.

Newly added paragraph (p 17):

I In addition to the confined CHB droplets discussed above, we verified that alginate colloidal ordering also emerges in a flattened CHB sessile drop geometry with a much larger interfacial radius of curvature (Supplementary Fig. 10a). Despite the altered geometry, the system exhibited a similar time scale for ordering: well-defined hexagonal domains appeared in the CHB phase approximately 8 hr after exposure to Ba^{2+} , consistent with the droplet-based experiments. This demonstrates that the ordering mechanism is robust and not restricted to the confined droplet environment, and that the sessile drop approach effectively illustrates the generality of the phenomenon. These observations also support the use of vial-based ζ -potential measurements to describe the electrostatic charging behavior of alginate hydrogel particles in both geometries.

(3) I presume alginate particles are soft and also deformable, what is the rationale for modeling interparticle interaction with Yukawa (screened-Coulomb) potential, which is typically used for hard colloids.

We thank the reviewer for raising this point. Although alginate hydrogel particles are mechanically soft and deformable, their electrostatic interactions in CHB are governed primarily by the charged surface layer rather than the elastic properties of the particles. In our system, the Debye screening length ($\kappa^{-1} \approx 2.5\text{--}3 \mu\text{m}$) is comparable to or larger than the particle radius, meaning that particles interact through long-range screened Coulomb repulsion at

[Type here]

separations far exceeding those at which mechanical deformation would occur. As a result, the softness of the hydrogel particles has negligible influence on the effective interparticle potential in the regime relevant to ordering. We have added clarification in the revised manuscript.

Before: For our colloidal system in CHB, the competition is between screened electrostatic potential energy and thermal kinetic energy. We model interparticle repulsion with a Yukawa (screened-Coulomb) potential $E_{\text{pot}} = \frac{4\pi\epsilon_0\epsilon\zeta^2a^2e^{2\kappa a}}{re^{\kappa r}}$, where a is the particle radius, r is the interparticle separation, ϵ_0 is the vacuum permittivity, and κ^{-1} is the Debye screening length.

After (p14): We define a dimensionless interaction parameter Γ to quantify the relative strength of electrostatic interactions compared to thermal fluctuations in our colloidal system dispersed in CHB. Interparticle repulsion is modeled using the Yukawa (screened Coulomb) potential $E_{\text{pot}} = \frac{4\pi\epsilon_0\epsilon\zeta^2a^2e^{2\kappa a}}{re^{\kappa r}}$, where a is the particle radius, r is the interparticle separation, ϵ_0 is the vacuum permittivity, and κ^{-1} is the Debye screening length. Although the alginate particles are mechanically soft, their interactions in CHB are dominated by the charged surface layer and occur at separations far larger than those at which mechanical deformation would be relevant. The Yukawa potential therefore provides an appropriate effective pair interaction for the present system.

(4) Authors should justify the need for using nested emulsions. This could have also been done by considering planar interfaces.

The nested-emulsion geometry was chosen because it provides a well-defined geometric confinement in which Ba^{2+} -mediated surface-charge buildup progressively strengthens electrostatic repulsion and ultimately drives the emergence of ordered structures. The microfluidic approach enables us to generate CHB droplets with well-controlled size and curvature radius, allowing systematic exploration of nonequilibrium ordering pathways while minimizing variability in interfacial geometry. Such control is difficult to achieve with a planar interface, where the local oil–water curvature against the container wall can vary between experiments. As we show in Supplementary Fig. 10, ordering also occurs in a flattened sessile-drop configuration, where the CHB–water interface has a much larger radius of curvature. This confirms that the ordering mechanism is robust and not restricted to the nested-emulsion environment. However, while the sessile drop method is useful for demonstrating the generality of the phenomenon, it does not provide the same level of reproducible curvature control as the microfluidic nested-emulsion system. For studying nonequilibrium systems and comparing data across multiple droplets, the nested-emulsion geometry therefore offers a more reliable and tunable experimental platform. We have added an explanation of these points in the revised manuscript to clarify the rationale for using the nested-emulsion configuration.

[Type here]

Before: Here, we developed a simple, room-temperature platform for studying Wigner-type crystallization: alginate hydrogel colloids confined within CHB droplets dispersed in an aqueous phase.

After (p3): Here, we develop a simple, room-temperature platform for studying time-dependent nonequilibrium ordering using soft alginate hydrogel colloids confined within CHB droplets dispersed in an aqueous phase. A microfluidic device produces alginate-nested emulsions with well-controlled droplet size and interfacial curvature, providing reproducible geometric confinement for monitoring ordering pathways.

Newly added statements (p17): In addition to the confined CHB droplets discussed above, we verified that alginate colloidal ordering also emerges in a flattened CHB sessile drop geometry with a much larger interfacial radius of curvature (Supplementary Fig. 10a). Despite the altered geometry, the system exhibited a similar time scale for ordering: well-defined hexagonal domains appeared in the CHB phase approximately 8 hr after exposure to Ba^{2+} , consistent with the droplet-based experiments. This demonstrates that the ordering mechanism is robust and not restricted to the confined droplet environment, and that the sessile drop approach effectively illustrates the generality of the phenomenon.

(5) Since the alginate particles are generated via microfluidic technique coupled with cross-linking, monodisperse particles are expected. However, authors discuss some of the results based on size segregation and particle size effect. This needs to be clarified.

We believe the confusion arises from mixing the roles of the two different droplet types in our nested-emulsion system. The microfluidic device produces monodisperse CHB droplets, but not the alginate droplets contained inside them. Before entering the microfluidic device, the alginate aqueous phase is mixed with CHB and dispersed by bath sonication, which creates a stochastically distributed population of alginate sub-droplets with inherently polydisperse sizes. This alginate-in-CHB dispersion is then supplied to the microfluidic device, which encapsulates these non-uniform alginate droplets within monodisperse CHB droplets. In addition, the alginate droplets undergo significant dynamical size evolution before and during Ba^{2+} diffusion and crosslinking. Prior to hydrogel stabilization, coalescence events occur, producing larger, unstable droplets. This early-time size evolution is directly visualized in the flattened CHB sessile drop geometry (Supplementary Fig. 10), where the mean diameter of alginate droplets initially increases within the first ~10 min and then progressively decreases over the next several hours as large, weakly crosslinked droplets rise to the CHB/water interface and merge into the external aqueous phase. Only the smaller droplets that remain in the CHB phase long enough to interact with diffusing Ba^{2+} ions become sufficiently crosslinked and form stable hydrogel particles. Because of these processes—stochastic droplet formation, early coalescence, and buoyancy-driven removal of larger droplets—the resulting alginate hydrogel particles inside each CHB droplet are necessarily polydisperse, and this polydispersity underlies the size-segregation effects discussed in the manuscript. We have clarified these points in the revised manuscript.

[Type here]

Before: The inner phase comprised sodium alginate droplets dispersed in CHB (dielectric constant $\epsilon \approx 7.9$) and the outer phase was ultrapure water containing 2 wt% poly(vinyl alcohol) (PVA) as a stabilizer. The resulting Alg@CHB emulsions (mean diameter $\sim 400 \mu\text{m}$) were collected and incubated in an aqueous barium acetate solution ($C_{\text{Ba}(\text{Ac})_2} = 40 \text{ mM}$ ($\sim 1 \text{ wt}\%$)), under which conditions ionically crosslinked alginate hydrogels formed, improving particle stability.

After (p5): The inner phase comprised sodium alginate droplets dispersed in CHB (dielectric constant $\epsilon \approx 7.9$), which were prepared by bath sonication of an alginate aqueous solution mixed with CHB, resulting in inherently polydisperse alginate sub-droplets. The outer phase was ultrapure water containing 2 wt% poly(vinyl alcohol) (PVA) as a stabilizer. The resulting Alg@CHB emulsions with a monodisperse size (mean diameter $\sim 400 \mu\text{m}$) were collected and incubated in an aqueous barium acetate solution ($C_{\text{Ba}(\text{Ac})_2} = 40 \text{ mM}$ ($\sim 1 \text{ wt}\%$)), under which conditions ionically crosslinked alginate hydrogels formed, improving particle stability.

Newly added paragraph (p17): To clearly visualize the ordering process, we monitored the edge region of the CHB sessile drop, where alginate dispersed structures beneath the CHB/water interface are optically well resolved (Supplementary Fig. 10b,c). Interestingly, the size of alginate droplets exhibited a characteristic transient evolution during the early stage (0–1 hr). The mean droplet size initially increased within the first 10 min and subsequently decreased over the next several hours (Supplementary Fig. 10d). We attribute this behavior to partial coalescence among weakly crosslinked alginate droplets, producing larger and unstable droplets that rise to the interface and merge into the external aqueous phase. Meanwhile, smaller alginate droplets remain in the CHB phase for longer, allowing sufficient contact with diffusing Ba^{2+} ions to build surface charge and form stable hydrogel particles that eventually assemble into ordered colloidal structures.

(6) As per Figure 5(d), zeta potential is positive whereas in Figure 2 in the supporting file, it is negative. This is contradicting reporting.

We thank the reviewer for pointing out this issue. The ζ -potential of alginate hydrogel particles is negative, as shown in Supplementary Fig. 2. Accordingly, the labels have been corrected to indicate the absolute value $|\zeta|$.

Revised Fig. 5:

[Type here]

Fig. 5 Estimation of the Debye screening length and the dimensionless interaction parameter. **a** Bright-field image (i) of an ordered lattice after 12 hr in 100 mM Ba(Ac)₂, used to extract the experimental spring constant k_s^{exp} (Supplementary Movie 1, recorded at 30 fps). Inset (ii) shows the full field of view with the analyzed region (yellow box); inset (iii) shows the probability density of the x -displacement u_x with a Gaussian fit, yielding $k_s^{\text{exp}} \approx 5.02 \text{ nN} \cdot \text{m}^{-1}$. **b** BD-predicted lattice spring constant k_s^{BD} as a function of zeta potential ζ for various κ^{-1} . The horizontal dashed line denotes k_s^{exp} . Error bars are standard deviations over 10 independent BD runs. **c** Contour map of the mismatch $\Delta k_s = |k_s^{\text{BD}} - k_s^{\text{exp}}|$ in the (ζ, κ^{-1}) plane. The white dashed curve marks $\Delta k_s \approx 0$. **d** Dimensionless interaction parameter Γ as a function of incubation time t for different Debye screening lengths κ^{-1} . The pink regions in panels **b–d** highlight the parameter band most consistent with experiment, constraining the Debye length to the few-micrometer range ($\kappa^{-1} \approx 2.5\text{--}3 \mu\text{m}$) and the zeta potential to around $\zeta \approx 50 \text{ mV}$.

[Type here]

Reviewer #2 (Remarks to the Author):

This study presents a clever and experimentally accessible colloidal analogue of Wigner crystallization of electrons but then using soft alginate particles dispersed in cyclohexyl bromide (CHB). By tuning electrostatic screening and monitoring lattice formation, the system reproduces key scaling features of electron Wigner crystals, notably the critical Wigner parameter ($\Gamma \approx 120\text{--}150$) at the onset of ordering. The consistent correspondence between measured and simulated lattice properties underscores the robustness of screened electrostatic interactions even in this nonequilibrium environment.

Overall, the work provides a compelling and well-constructed analogy between colloidal and electronic Wigner crystallization. Conceptually, it succeeds in showing how a universal organizing principle—the dominance of electrostatic repulsion over thermal motion—governs lattice formation across very different physical systems. Quantitatively, the study demonstrates impressive internal consistency: the experimentally determined Wigner parameter and screening length coincide with both simulations and prior literature, lending credibility to the physical interpretation. The experimental platform itself is another strength—simple, tunable, and compatible with real-time imaging—offering a practical means to study charge-driven self-assembly.

We sincerely thank the reviewer for the thoughtful and encouraging assessment of our work. We are grateful that the reviewer recognizes the strength of our experimental platform and the internal consistency between experiments and simulations.

Nonetheless, several limitations remain. The simulations omit polydispersity and interaction heterogeneity, which likely overstate lattice stability; the physical origin and temporal stability of screening in CHB remain uncertain (see further); and the nonequilibrium nature of the system introduces reproducibility challenges. Looking forward, the framework holds significant promise, especially if complemented by direct measurements of ionic dissociation and particle interactions. Such refinements could transform this elegant soft-matter analogue into a quantitatively reliable model for exploring Wigner-type ordering and melting phenomena.

We appreciate the reviewer's insightful comments regarding the limitations of our current framework—particularly the omission of polydispersity and interaction heterogeneity in simulations, as well as the complexity of electrostatic stability in CHB. We have revised the manuscript to address them more explicitly.

Regarding the comments on simulation, we agree that assuming monodisperse particles and homogeneous interactions can overestimate lattice stability. Indeed, in our previous work we reported that interaction heterogeneity promotes melting in Monte Carlo simulations [Soft Matter 6, 5327 (2010); ACS Appl. Polym. Mater. 2, 1304 (2020)]. To address this point, we performed additional BD simulations incorporating the experimentally observed particle-size distribution and analyzed the resulting effects on the microstructures and the lattice spring constant. We found that increasing particle-size polydispersity induces melting-like structural disorder—consistent with the prior studies—while the mean lattice spring constant remains

[Type here]

largely insensitive to polydispersity. These findings have been added to the revised manuscript.

Newly added paragraph (p14): To assess the robustness of the spring-constant-based calibration, we additionally examined how particle-size polydispersity (i.e., the coefficient of variation, CV) influences the simulated lattice mechanics. As detailed in Supplementary Note 3, we introduced experimentally relevant size variations by sampling particle radii from a Gamma distribution^{37,38} and performed BD simulations at matched surface coverage. Increasing polydispersity reproduces the expected melting-like microstructural disorder (Supplementary Fig. 9a–i), consistent with prior studies,^{37,38} yet the resulting lattice spring constant remains largely insensitive to CV over the range tested (Supplementary Fig. 9j). This insensitivity indicates that the lattice spring constant predominantly reflects the underlying interaction scale rather than the degree of polydispersity, supporting the reliability of the k_s -based screening-length calibration used in Fig. 5b,c.

Before: The same screening length ($\kappa^{-1} \approx 3 \mu\text{m}$) also produced the best agreement between simulated and experimental lattice spring constants. While our BD simulations did not explicitly include interaction heterogeneity or polydispersity—both known to promote melting in colloids^{47,48}—the consistent estimate of $\kappa^{-1} \approx 3 \mu\text{m}$ within the experimentally relevant $\zeta \sim 50\text{--}60 \text{ mV}$ range indicates that finite electrostatic screening is robust in this nonequilibrium environment.

After (p17): By calibrating BD simulations to the experimentally measured lattice spring constant, we estimate the Debye screening length as $\kappa^{-1} \approx 2.5\text{--}3 \mu\text{m}$ in the alginate colloid system. This estimate remains robust when particle-size polydispersity is incorporated into the simulations, indicating that the spring constant primarily reflects the underlying interaction scale rather than size heterogeneity. The inferred screening length also falls within the range reported previously for colloidal systems in low-dielectric media.^{18,33–36}

Newly added Supplementary Fig. 9 (pS10 in SI):

[Type here]

Supplementary Fig. 9 Effects of particle-size polydispersity on microstructure and lattice spring constant in BD simulations. **a-d** Delaunay triangulations of simulated colloidal lattices for varying particle-size polydispersity, quantified by the coefficient of variation (CV): $CV = 0$ (a), 0.13 (b), 0.2 (c), 0.3 (d). Colors indicate distinct coordination environments. **e-h** Corresponding histograms of the coordination number for each CV condition, showing broadening distributions as polydispersity increases. **i** Radial distribution function (rdf) plotted as a function of normalized separation $r \cdot \langle R \rangle^{-1}$, illustrating the progressive attenuation of peak amplitudes and the loss of long-range order as CV increases. **j** Mean lattice spring constant (k_s^{BD}) extracted from BD simulations as a function of CV , showing that k_s^{BD} remains largely insensitive to particle-size polydispersity within the range tested. Error bars represent the standard deviation over ten independent simulation runs.

Newly added Supplementary Note 3 (pS10 in SI):

Supplementary Note 3. Implementation of particle-size polydispersity in BD simulations

To examine how particle-size polydispersity affects the microstructure and the effective lattice spring constant (k_s) of the colloidal assembly structure, we performed BD simulations with $N = 100$ particles interacting via a screened-Coulomb (Yukawa) potential. The box size was chosen such that the surface coverage matched that used in Fig. 5a. Experimentally, the top-layer alginate hydrogel particles in Fig. 5a exhibit a mean radius $\langle R \rangle_{\text{exp}} \approx 1.42 \mu\text{m}$ and a standard deviation $\sigma_{R,\text{exp}} \approx 0.19 \mu\text{m}$, corresponding to a coefficient of variation, $CV_{\text{exp}} = \sigma_{R,\text{exp}} / \langle R \rangle_{\text{exp}} \approx 0.13$. To implement controlled polydispersity while ensuring strictly positive radii, we sampled particle sizes from a Gamma distribution, $R \sim \Gamma(k, \theta)^{3,4}$ where the shape parameter k and scale parameter θ satisfy a mean $\mu = \langle R \rangle = k\theta$ and a variance $\sigma^2 = k\theta^2$. The coefficient of variation (CV) is $CV = \frac{\sigma}{\mu} = \frac{1}{\sqrt{k}}$, which gives $k = \frac{1}{CV^2}$ and $\theta = CV^2 \mu$.

[Type here]

In simulations, we set the target mean radius to $R_0 = 1.42 \mu\text{m}$ and varied the imposed CV . We then sampled $R_i \sim \Gamma(k, \theta)$, $i = 1, \dots, N$.

Particles were initialized on a slightly perturbed square lattice in a periodic box. Each particle experienced overdamped Brownian dynamics with drag coefficient $\gamma_i = 6\pi\eta R_i$ and diffusion coefficient $D_i = \frac{k_B T}{\gamma_i}$. The pairwise Yukawa interaction between particles i and j at separation r_{ij} had effective amplitude, $U_{0,ij} = 4\pi\epsilon_0\epsilon\zeta^2\kappa R_i R_j e^{\kappa(R_i+R_j)}$, and force $F_{ij}(r_{ij}) = U_{0,ij} \frac{e^{-\kappa r_{ij}}}{\kappa r_{ij}^2} (1 + \kappa r_{ij})$. After an equilibration period, particle trajectories were recorded at fixed time intervals.

As shown in Supplementary Fig. 9a–h, Delaunay triangulations and coordination-number histograms show that increasing CV progressively reduces the fraction of sixfold-coordinated sites, giving rise to defect-rich, melted-like configurations. This trend is consistent with our earlier Monte Carlo results, demonstrating that interaction heterogeneity promotes melting and destabilizes long-range order.^{3,4} Likewise, the radial distribution function (rdf) becomes increasingly attenuated and broadened with larger CV , reflecting a clear loss of long-range order (Supplementary Fig. 9i). In contrast to the strong microstructural degradation, the effective lattice spring constant k_s shows virtually no dependence on CV across the range studied. The mean values from ten independent runs overlap substantially (Supplementary Fig. 9j), indicating that the averaged local confining stiffness is relatively insensitive to particle-size polydispersity despite pronounced structural disorder. These results support the use of k_s as a reliable measure of the interaction scale in our colloidal assemblies and validate the interpretation of Fig. 5 in the main text.

A detailed response to the reviewer's comments on CHB impurities and screening stability is provided in our reply to the following comment.

We have also expanded the Discussion section to address the incomplete understanding of the driving forces and transport pathways governing Ba^{2+} transfer through the CHB phase, and we highlight future directions involving direct interparticle force measurements using optical laser tweezers.

Before: Looking ahead, microscale temperature control could probe thermal stability, dilute-condition optical-tweezer measurements of interparticle repulsion could refine the interaction model and clarify melting pathways. Such studies will further bridge colloidal and electronic Wigner crystals and broaden our understanding of charge-driven self-organization in nonequilibrium settings.

[Type here]

After (p18): Looking ahead, several directions could further deepen the mechanistic understanding of the observed ordering. Incorporating single-particle dynamical analyses, such as mean-squared displacements and cage escape times, would provide direct insight into relaxation pathways beyond the structural evolution emphasized here. Microscale temperature control could enable systematic exploration of thermal stability, while dilute-condition direct-force measurements between alginate hydrogels—such as optical tweezer-based interaction profiling^{4,18}—could further refine interaction models and clarify melting pathways. In addition, although Ba^{2+} ions clearly penetrate the CHB medium to crosslink alginate droplets, the microscopic mechanisms governing this ion transfer remain incompletely understood. Future studies quantifying interfacial partitioning, reaction–diffusion kinetics, and counterion exchange will be essential for determining how charge buildup and screening evolve in weakly dissociative media. Together, these efforts would provide a deeper mechanistic foundation for electrostatically mediated colloidal ordering under nonequilibrium conditions and broaden the framework for understanding charge-regulated assembly in low-dielectric environments.

One key concern regards the solvent used. CHB is only weakly dissociative, and its ionic content—and hence the effective screening length ($\kappa^{-1} \approx 3 \mu\text{m}$)—depends sensitively on trace impurities, counterion exchange, and residual water or surface active component contamination and possibly time. The chemical stability and reproducibility of the electrostatic environment with CHB has been discussed in literature for colloidal crystals of PMMA, and the properties of CHB which may evolve over hours as ions partition between the alginate network and the solvent and batch to batch differences between CHB solvents. While the framework offers a practical route to visualize charge-driven ordering, a clearer understanding of ion dissociation and mobility in CHB should be discussed.

We appreciate the reviewer’s comments regarding the chemical stability of CHB and its influence on electrostatic screening. We agree that CHB is only weakly dissociative and that its ionic content can be sensitive to trace impurities, counterion exchange, and residual water. In fact, our earlier work on PMMA colloids in CHB/n-decane mixtures directly demonstrated that interparticle forces in CHB are highly responsive to such impurities [Soft Matter 15, 8051 (2019)]. In that study, we showed that (i) UV-curable adhesives commonly used for flow-cell fabrication can partially dissolve into CHB, significantly reducing measured electrostatic forces, and (ii) unfiltered CHB contains some impurities that strongly screen repulsion, with forces nearly 80% lower than those measured after filtering CHB through activated alumina. These results highlight the importance of carefully controlling solvent purity in CHB-based electrostatic experiments. For the present work, we followed these guidelines strictly. We used freshly filtered CHB for all experiments and ensured that CHB droplets or sessile drops did not contact any UV-curable adhesive. Although minor ionic drift in CHB cannot be entirely ruled

[Type here]

out, the dominant ion-exchange pathway in our system is intentional: Ba^{2+} ions from the external aqueous phase diffuse through CHB and crosslink alginate droplets into hydrogels. This Ba^{2+} influx is the primary driver of the time-dependent surface-charge buildup that strengthens repulsive interactions and induces alginate colloidal ordering. Moreover, if CHB instability or impurity-induced screening were the dominant effect, one would expect a decrease in repulsive interactions over time. Instead, across both the nested-emulsion geometry (Fig. 2) and the flattened sessile drop geometry (Supplementary Fig. 10), we observe that the interparticle interactions increase and that the ordered structures remain stable over the relevant timescale. This indicates that impurity-related screening effects are negligible compared with the Ba^{2+} -induced surface-charge buildup. We have incorporated these clarifications into the revised manuscript.

Before: After confirming stable production for ~5–7 min, the CHB droplets were collected into a Petri dish containing the desired ion solution.

After (p20): After confirming stable production for ~5–7 min, the CHB droplets were collected into a cylindrical chamber containing the desired ion solution. The chamber was constructed by bonding a glass ring to a circular coverslip (Marienfeld, No. 1.5H, Germany) using a UV-curable adhesive (Optical Adhesive 81, Norland, USA). The assembled chamber was thoroughly cleaned with ethanol and DI water, followed by oxygen plasma treatment for 30 s using a plasma cleaner (PDC-32G-2, Harrick Plasma, USA). Because the collected CHB droplets are dispersed in the aqueous phase, they do not come into contact with the UV adhesive, thereby eliminating the possibility of impurity generation in CHB through adhesive dissolution.¹⁸

Newly added statements (p23): Importantly, the CHB sessile drop remained localized on the hydrophobic central region and did not reach the adhesive-sealed boundary, ensuring that it never contacted the UV-curable adhesive and eliminating the possibility of impurity generation through adhesive dissolution.¹⁸

[Type here]

Reviewer #3 (Remarks to the Author):

The manuscript by Jung et al. reports the formation of ordered hexagonal patterns in charged alginate hydrogel microparticles confined within CHB droplets, which the authors interpret as a room-temperature “model Wigner-type crystal.” The system shows interesting colloidal self-assembly: Ba²⁺-mediated crosslinking increases repulsion over time, buoyancy-driven compaction yields quasi-2D layering, and hexagonal order is quantified via Voronoi/FFT metrics. They also show reversible disorder–order after magnetic/mechanical perturbations. However, I have major concerns about the central premise and interpretation, leading me to recommend REJECT. The work is framed as a “macroscopic, real-time analogue” of electronic Wigner crystallization and draws explicit parallels to electron systems. This analogy is not physically justified: it conflates interaction-driven localization of mobile carriers with equilibrium self-assembly of intrinsically localized charged objects. These are not issues that can be addressed through revision, they reflect a fundamental mischaracterization of physics that would mislead readers about the nature of Wigner crystallization. A Wigner crystal, by definition, arises when free, mobile charge carriers (electrons) become spatially ordered because Coulomb repulsion energy overwhelms their kinetic energy—the very kinetic energy that would otherwise allow them to remain itinerant. Classic realizations—electrons trapped on liquid helium surfaces (Grimes & Adams, *Phys. Rev. Lett.* 42, 795, 1979) or magnetically induced Wigner solids in ultra-clean two-dimensional electron gases (Andrei et al., *Phys. Rev. Lett.* 60, 2765, 1988)—involve no underlying lattice and no external forces. Ordering emerges purely from the competition between interaction and kinetic energies of the carriers themselves. The present system bears no resemblance to this physics. The charges are intrinsically localized in crosslinked polyelectrolyte droplets from the outset. Just like apples in a box. There are no mobile carriers, no electronic band structure, no hopping transport, and no itinerant kinetic energy to overcome. The system forms a nonequilibrium colloidal crystal under time-dependent charging (progressive Ba²⁺ crosslinking) and external forces (buoyancy-driven stratification), not through a thermodynamic phase transition driven by interaction-kinetic energy competition. What we observe is simply a colloidal Yukawa lattice—well-studied soft-matter self-assembly—where pre-localized objects arrange under repulsive interactions and gravity. Any insulator has localized electrons, this does not make every ordered insulating structure a Wigner crystal.

We thank the reviewer for the careful reading of our manuscript and for acknowledging the experimental observations of colloidal self-assembly, including the time-dependent strengthening of repulsion, buoyancy-driven layering, the characterization of hexagonal order, and the reversible disorder–order transitions under external perturbations. We agree that the primary issue raised here concerns the central premise and interpretation of the work rather than the experimental execution itself.

We fully agree with the reviewer that our system should not be framed as a macroscopic, real-time analogue of electronic Wigner crystallization, and that a direct correspondence to quantum electronic systems is not physically justified. In particular, electronic Wigner crystals arise from the localization of mobile charge carriers governed by the competition between long-range Coulomb interactions and quantum kinetic energy, whereas our system consists of

[Type here]

intrinsically localized colloidal objects undergoing interaction-driven assembly under screened electrostatic interactions and thermal motion. This distinction was also explicitly emphasized by Reviewer #4, who noted that the analogy between electronic Wigner crystallization and colloidal ordering is, at best, heuristic, and that the two systems are governed by fundamentally different physics. We fully concur with this assessment.

At the same time, we note that the term Wigner-type (or Wigner-like) crystallization has been widely used in the soft-matter literature to describe classical colloidal systems in which long-range repulsive interactions dominate over thermal fluctuations, leading to ordered structures [Phys. Rev. E 91, 032310 (2015); Proc. Natl. Acad. Sci. 104, 2585-2590 (2007); Nat. Commun. 5, 4049 (2014); Phys. Chem. Chem. Phys. 18, 5211-5218 (2016)]. In this established context, the terminology refers to a shared organizing principle rather than to a direct mapping of microscopic degrees of freedom, transport mechanisms, or quantum physics.

To avoid conceptual ambiguity, we have substantially revised the manuscript. We have removed or revised all parts that framed our system as a direct analogue of electronic Wigner crystallization. Importantly, the main experimental results and quantitative analyses remain unchanged. The revised manuscript now focuses on presenting these results as a study of interaction-driven ordering in a nonequilibrium colloidal system under screened electrostatic interactions, time-dependent charging, and quantitative analysis supported by BD simulations. We believe that this reframing addresses the reviewer's concern regarding the central premise and provides a clearer and more accurate interpretation of the experimental findings.

Before: Dynamics of Nonequilibrium Colloidal Wigner-Type Crystals

After (title): Dynamics of Nonequilibrium Colloidal Ordering Driven by Electrostatic Evolution

Before: Wigner crystals, highly ordered lattices of repelling charges, are typically only accessible under extreme conditions, limiting direct experimental study. Here we introduce a simple, tunable, room-temperature model system: alginate hydrogel colloids confined within cyclohexyl bromide (CHB) droplets dispersed in an aqueous phase. Ion-induced crosslinking generates soft, charged alginate particles whose repulsion increases over time as surface charge builds up, while buoyancy-driven compaction forms a stack of locally quasi-2D layers within CHB. The ensemble spontaneously orders into a triangular lattice that persists as a nonequilibrium solid state. Quantitative image analysis identifies crystallization onset when the dimensionless Wigner parameter of Γ —the mean electrostatic-to-thermal energy ratio—reaches ≈ 117 – 149 , comparable to reported critical values for screened electron systems ($\Gamma_c \approx 95$ – 150). Brownian dynamics simulations reproduce the ordering and indicate a Debye screening length of $\sim 3 \mu\text{m}$ under experimental conditions. The colloidal Wigner crystals show reversible ordering under mechanical and magnetic perturbations, providing a simple, real-time platform to study charge-mediated self-organization and the physics of Wigner crystallization under nonequilibrium conditions.

[Type here]

After (Abstract): Elucidating how repulsive interactions evolve to generate ordered structures in nonequilibrium colloidal systems remains a central challenge, owing to the scarcity of experimental platforms that provide particle-resolved, real-time access to their structural evolution. Here we introduce a simple, tunable, room-temperature model in which alginate hydrogel colloids confined within cyclohexyl bromide (CHB) emulsion droplets spontaneously organize under evolving electrostatic interactions. Ba^{2+} ions diffusing from the surrounding aqueous phase progressively crosslink the alginate droplets, increasing their surface charge, while buoyancy compacts them into locally quasi-two-dimensional layers within the CHB phase. As electrostatic repulsion strengthens, the assembly evolves from a disordered state to a triangularly ordered structure. By calibrating Brownian dynamics simulations to the experimentally measured lattice spring constant, we constrain the effective Debye screening length to approximately 2.5–3 μm . Quantitative imaging further reveals that ordering emerges once the dimensionless interaction parameter—defined as the ratio of electrostatic to thermal energy—reaches values of approximately 117–149. The ordered state exhibits reversible disordered–order behavior under mechanical and magnetic perturbations, demonstrating a robust nonequilibrium platform for probing charge-regulated colloidal ordering under confinement.

Fully revised Introduction (p3):

Colloidal systems provide a versatile model platform for studying interaction-driven ordering at experimentally accessible length and time scales. Particle-resolved imaging, tunable interactions, and slow relaxation dynamics enable real-time visualization of ordering pathways, offering insights that complement those from atomic and electronic systems.¹⁻⁷ In low-dielectric media such as cyclohexyl bromide (CHB)–decalin mixtures, long-range screened electrostatic repulsion under weak screening can stabilize ordered colloidal phases even at low particle volume fraction.⁸⁻¹⁰ Such systems exhibit rich behaviors, including re-entrant melting, glass formation, and tunable elasticity, which are well described by established theoretical frameworks such as Yukawa, one-component-plasma, and charge-regulation models.^{11, 12} Despite these advances, most prior studies have focused on equilibrium structures, whereas the time-dependent mechanisms governing surface-charge buildup, evolving screening length, and ordering under confinement have not been extensively explored.

Interaction-driven ordering in colloidal systems generally arises when electrostatic interactions become comparable to or exceed thermal fluctuations, leading to the emergence of ordered structures. In the soft-matter literature, this regime is sometimes described using the term Wigner-type or Wigner-like ordering,⁹⁻¹² reflecting its historical origin in the concept of Wigner crystallization, in which ordered electronic states emerge from the competition between long-range Coulomb interactions and kinetic energy at low carrier densities.¹³⁻¹⁷ In the colloidal assemblies, this terminology is used to denote a regime in which electrostatic interactions dominate over thermal fluctuations and give rise to ordered structures. In the present work, we focus on this interaction-dominated ordering behavior in colloidal particles under screened electrostatic interactions.

Here, we develop a simple, room-temperature platform for studying time-dependent nonequilibrium ordering using soft alginate hydrogel colloids confined within CHB droplets dispersed in an aqueous phase. A microfluidic device produces alginate-nested emulsions with well-controlled droplet size and interfacial curvature, providing reproducible geometric

[Type here]

confinement for monitoring ordering pathways. CHB, with its low-dielectric constant, supports strong electrostatic repulsion under weak screening,^{10, 18} while divalent cations (e.g., Ba²⁺) diffusing from the aqueous phase progressively crosslink the alginate droplets, gradually increasing their surface charge. Buoyancy-driven stratification then yields quasi-two-dimensional (2D) layers that spontaneously develop triangular order. As a result of this ion-mediated charging process, the strength of interparticle electrostatic interactions evolves continuously in time. In this context, nonequilibrium refers to the temporal evolution of interparticle interaction strength. Although particle configurations can relax into quasi-steady structural states on relatively short timescales, the interaction strength evolves more slowly, rendering the overall ordering process intrinsically nonequilibrium. We quantify the onset and kinetics of this ordering using particle-resolved imaging, Voronoi/Delaunay analyses, and fast Fourier transform (FFT)-based metrics. To establish a quantitative interaction scale, we calibrate Brownian dynamics (BD) simulations to the experimentally measured lattice spring constant, enabling estimation of the Debye screening length under our experimental conditions. We then characterize the time-dependent strengthening of electrostatic interactions using a dimensionless interaction parameter Γ , defined as the ratio of electrostatic energy to thermal energy. We further map layer-resolved structural ordering and examine the effects of ion valence and ion concentration. Finally, we demonstrate reversible disorder-order transitions under magnetic and mechanical perturbations, together establishing a controllable nonequilibrium platform for studying classical, interaction-driven colloidal ordering.

The authors' definition of the Wigner parameter Γ also exemplifies the conceptual confusion. They define Γ as the ratio of Yukawa interaction energy to Brownian thermal energy of colloidal particles. In electronic Wigner physics, the relevant energy scale is electronic kinetic energy—Fermi energy for quantum systems, or kinetic energy of mobile point charges in classical plasmas. Equating Brownian fluctuations of permanently localized droplets with electronic kinetic energy is invalid. The authors note their measured $\Gamma \sim 117\text{--}149$ matches electronic Wigner crystal thresholds ($\Gamma \sim 95\text{--}150$) and present this as evidence, but numerical coincidence means nothing when the underlying degrees of freedom, conservation laws, and dynamical processes differ categorically. The authors claim their platform enables observation of "electronic behavior at macroscopic scale," but Wigner crystallization physics does not scale trivially with particle size. Quantum mechanics itself cannot be reproduced by enlarging particles and substituting thermal energy for electronic kinetic energy. The authors provide no rigorous dimensional analysis or dynamical mapping to justify that their system preserves the relevant physics. While the experimental observations are competently executed, the central claim—that this system realizes or analogizes electronic Wigner crystallization—is not physically justified. The distinction between interaction-driven localization of mobile carriers and structural self-assembly of fixed charged objects is fundamental physics, not interpretation.

We thank the reviewer for the comment regarding the definition of the Wigner parameter Γ . We agree that the energy scales entering Γ in electronic Wigner systems and in colloidal assemblies are fundamentally different. Accordingly, we have revised manuscript to clarify the role of Γ in the present context. In the revised manuscript, Γ is defined as a dimensionless interaction parameter within the soft-matter framework, used to quantify the dominance of interparticle repulsion over thermal positional fluctuations in a Yukawa-type colloidal system.

[Type here]

In practice, Γ is employed to track the time-dependent evolution of interaction strength as electrostatic repulsion develops and to provide a dimensionless reference for identifying the onset of colloidal ordering. Reflecting this clarification, we have reorganized the relevant section of the manuscript. We believe that these revisions resolve the conceptual issue raised by the reviewer while retaining Γ as a useful descriptive parameter for interaction-driven ordering in colloidal systems.

Revised Fig. 5 (p12):

Fig. 5 Estimation of the Debye screening length and the dimensionless interaction parameter. **a** Bright-field image (i) of an ordered lattice after 12 hr in 100 mM Ba(Ac)₂, used to extract the experimental spring constant k_s^{exp} (Supplementary Movie 1, recorded at 30 fps). Inset (ii) shows the full field of view with the analyzed region (yellow box); inset (iii) shows the probability density of the x -displacement u_x with a Gaussian fit, yielding $k_s^{\text{exp}} \approx 5.02$ nN·m⁻¹. **b** BD-predicted lattice spring constant k_s^{BD} as a function of zeta potential ζ for various κ^{-1} . The horizontal dashed line denotes k_s^{exp} . Error bars are standard deviations over 10 independent BD runs. **c** Contour map of the mismatch $\Delta k_s = |k_s^{\text{BD}} - k_s^{\text{exp}}|$ in the (ζ, κ^{-1}) plane. The white dashed curve marks $\Delta k_s \approx 0$. **d** Dimensionless interaction parameter Γ as a function of incubation time t for different Debye screening lengths κ^{-1} . The pink regions in panels **b–d** highlight the parameter band most consistent with experiment, constraining the Debye length to the few-micrometer range ($\kappa^{-1} \approx 2.5\text{--}3$ μm) and the zeta potential to around $\zeta \approx 50$ mV.

[Type here]

Revised section (p12):

Estimation of Debye screening length

To estimate the Debye screening length κ^{-1} , we calibrated BD simulations to the experimentally measured lattice spring constant (k_s^{exp}) extracted from cage-relative thermal displacements (Fig. 5a and supplementary Movie 1).²⁹⁻³¹ The spring constant estimator was independently validated by simulating a single particle in a known 2D harmonic trap and recovering the input stiffness (see Supplementary Fig. 7 and Supplementary Note 2). Here the spring constant serves as a quantitative descriptor of the local stiffness experienced by each particle within the ordered colloidal lattice, reflecting the degree of electrostatic confinement imposed by neighboring particles.³²

Using a Yukawa potential (Supplementary Fig. 8), we performed BD simulations in which both the zeta potential ζ and the screening length κ^{-1} were systematically and independently varied to obtain the lattice spring constant $k_s^{\text{BD}}(\zeta, \kappa^{-1})$ (Fig. 5b). Comparison of this simulated parameter space with the experimentally measured k_s^{exp} yields a mismatch map, $\Delta k_s = |k_s^{\text{BD}} - k_s^{\text{exp}}|$ (Fig. 5c), from which the best agreement ($|\Delta k_s| \approx 0$) is identified. The corresponding region (pink shading in Fig. 5b,c) lies within an experimentally reasonable range of $\zeta \approx 50\text{--}60$ mV (Supplementary Fig. 2) and yields an effective Debye screening length of $\kappa^{-1} \approx 2.5\text{--}3$ μm .

Because the ordering process evolves under conditions of continuously changing interaction strength, assigning unique values of ζ and κ^{-1} at any specific time is inherently difficult. Nevertheless, by assuming a representative $\zeta \approx 50$ mV, the BD–experiment matching yields an effective screening length of $\kappa^{-1} \approx 3$ μm . This estimate is consistent with our prior optical tweezer measurements on poly(methyl methacrylate) (PMMA) particles in CHB/n-decane mixtures, which reported $\kappa^{-1} \approx 2.5$ μm .¹⁸ More broadly, reported screening lengths in nonpolar or weakly polar media fall in the micrometer range—for example, ~ 1 μm for PMMA particles in cis-decalin/CHB with poly-12-hydroxystearic acid,³³ $\sim 0.5\text{--}12$ μm for polystyrene or PMMA particles in hexadecane with cationic surfactant (sodium di-2-ethyl-hexylsulfosuccinate),^{34,35} and $\sim 0.8\text{--}8$ μm for PMMA in hexane with sorbitan trioleate.³⁶

Revised section (p14):

Dimensionless interaction parameter and time-dependent ordering

We define a dimensionless interaction parameter Γ to quantify the relative strength of electrostatic interactions compared to thermal fluctuations in our colloidal system dispersed in CHB. Interparticle repulsion is modeled using the Yukawa (screened Coulomb) potential

$E_{\text{pot}} = \frac{4\pi\epsilon_0\epsilon\zeta^2 a^2 e^{2\kappa a}}{r e^{\kappa r}}$, where a is the particle radius, r is the interparticle separation, ϵ_0 is the vacuum permittivity, and κ^{-1} is the Debye screening length. Although the alginate particles are mechanically soft, their interactions in CHB are dominated by the charged surface layer and occur at separations far larger than those at which mechanical deformation would be relevant. The Yukawa potential therefore provides an appropriate effective pair interaction for the present system. Using the thermal energy $k_B T$ as the reference energy scale, the interaction

parameter is defined as $\Gamma = \frac{E_{\text{pot}}}{k_B T} = \frac{\pi\epsilon_0\epsilon\zeta^2 \langle D \rangle^2 e^{\kappa \langle D \rangle}}{k_B T \langle r_{cc} \rangle e^{\kappa \langle r_{cc} \rangle}}$, using experimentally measured values of $\langle D \rangle$, $\langle r_{cc} \rangle$, and ζ (Fig. 2e and Supplementary Fig. 2). Low Γ corresponds to weakly correlated, fluid-like configurations, whereas large Γ indicates increasingly interaction-dominated,

[Type here]

ordered colloidal states.

Fig. 5d shows that Γ increases with incubation time and with increasing screening length, reflecting the progressive strengthening of electrostatic repulsions. A slight dip in Γ at shorter screening lengths can be attributed to the lower $\langle r_{cc} \rangle$ values observed in the disordered state. Considering the range of $\kappa^{-1} \approx 2.5\text{--}3\ \mu\text{m}$ estimated from the BD analysis (Fig. 5b,c), Γ reaches values of $\sim 117\text{--}149$ at $t \approx 6$ hr, coinciding with the experimentally observed onset of hexagonal ordering (Fig. 2). At later times ($t \gtrsim 8$ hr), Γ exceeds ~ 220 (pink regions in Fig. 5d), corresponding to a strongly interaction-dominated regime in which robust colloidal ordering is established. Although the microscopic mechanisms underlying electronic Wigner crystallization are fundamentally different from those governing colloidal ordering, the interaction parameter values at which stable ordering emerges in our colloidal system are of similar numerical magnitude to values often quoted for screened electronic Wigner systems ($\sim 95\text{--}150$).^{14-16,39} Here, this numerical correspondence is not interpreted as evidence of physical equivalence, but rather as providing a useful reference point for identifying the boundary between weakly correlated and strongly ordered regimes in colloidal assembly systems.

Revised paragraph (p18):

Building on this interaction length scale, we use the dimensionless interaction parameter Γ to summarize the time-dependent strengthening of electrostatic interactions during ordering. As surface charge builds up through ion diffusion and alginate hydrogelation, the resulting electrostatic repulsion strengthens, leading to an increase in Γ with incubation time. Γ reaches values of $\sim 117\text{--}149$ at $t \approx 6$ hr for $\kappa^{-1} \approx 2.5\text{--}3\ \mu\text{m}$, coinciding with the experimentally observed onset of triangular ordering. At later times ($t \gtrsim 8$ hr), Γ exceeds ~ 220 , corresponding to a strongly interaction-dominated regime in which robust colloidal ordering is established. At each incubation time, particle configurations appear to relax on timescales much shorter than those associated with surface-charge buildup and screening evolution, allowing the system to reach quasi-steady structural states despite the continuous change in interaction strength. This separation of timescales justifies the use of equilibrium-based structural descriptors—such as the lattice spring constant and Γ —at each stage, even though the overall ordering process remains intrinsically nonequilibrium.

The authors should consider rephrasing this work reframed as a study of colloidal crystal formation under screened electrostatic interactions, time-dependent charging, and geometric confinement, without invoking the Wigner crystal analogy their system cannot support.

We thank the reviewer for this constructive suggestion. We agree with this assessment and have substantially reframed the manuscript accordingly. In the revised version, the work is now presented explicitly as a study of colloidal crystal formation under screened electrostatic interactions, time-dependent charging, and quantitative analysis supported by BD simulations. Within this framework, the system provides a well-controlled experimental platform for probing interaction-driven ordering in nonequilibrium colloidal assemblies and for systematically tracking how electrostatic interactions and confinement influence structural evolution over time. We believe that this reframing more accurately reflects the physical scope of the system and strengthens the clarity and focus of the manuscript.

[Type here]

Reviewer #4 (Remarks to the Author):

The manuscript by Jung et al. presents a novel experimental system for studying the formation of two-dimensional ordered colloidal arrays. The approach, involving in-situ crosslinking of alginate hydrogels within low-dielectric CHB droplets, offers an accessible, room-temperature platform for investigating “Wigner-type” crystallization. The experiments are carefully designed, and the resulting structures are thoroughly analyzed, with corroboration from Brownian dynamics simulations. Overall, the manuscript represents a valuable contribution to the field.

We thank the reviewer for the positive and encouraging evaluation of our work. We appreciate the reviewer’s recognition of the novelty of our experimental system, the careful design of the measurements, and the consistency between our structural analyses and BD simulations. We are grateful for the reviewer’s thoughtful comments, which helped us further refine and clarify the manuscript.

My main concern lies with the direct analogy drawn to the electronic Wigner crystal in quantum systems. In my view, this analogy is at best heuristic. As the authors themselves note in the section “Estimation of Wigner parameter and Debye screening length”, the quantum phase transition is governed by the competition between long-range Coulomb forces and quantum kinetic energy, whereas the ordering observed here arises from the interplay of screened Yukawa potentials and thermal energy. This is a fundamental distinction that should be clearly articulated early in the Introduction. Even towards the end in the Discussion (starting line 349), the phrasing tends to blur this difference. A systematic study of colloidal Wigner crystallization is important and stands on its own merit; however, the manuscript would benefit from a clear and consistent separation between the colloidal analogue and the quantum electronic case. If the authors intend to maintain a stronger equivalence, some direct experimental evidence would be necessary to justify the claim.

We fully agree that the analogy between electronic Wigner crystallization and the colloidal ordering studied here is heuristic rather than literal, and that the underlying physics of the two systems is fundamentally different. As the reviewer correctly notes, electronic Wigner crystallization is governed by the competition between long-range Coulomb interactions and quantum kinetic energy, whereas the ordering observed in our system arises from the interplay between screened (Yukawa-type) electrostatic interactions and thermal motion in a classical colloidal environment. Accordingly, we have substantially revised the manuscript to clearly and consistently articulate this distinction. We believe that the revised framing addresses the reviewer’s concern and results in a clearer and more accurate presentation of the scope and implications of our work.

Before: Dynamics of Nonequilibrium Colloidal Wigner-Type Crystals

After (title): Dynamics of Nonequilibrium Colloidal Ordering Driven by Electrostatic Evolution

[Type here]

Before: Wigner crystals, highly ordered lattices of repelling charges, are typically only accessible under extreme conditions, limiting direct experimental study. Here we introduce a simple, tunable, room-temperature model system: alginate hydrogel colloids confined within cyclohexyl bromide (CHB) droplets dispersed in an aqueous phase. Ion-induced crosslinking generates soft, charged alginate particles whose repulsion increases over time as surface charge builds up, while buoyancy-driven compaction forms a stack of locally quasi-2D layers within CHB. The ensemble spontaneously orders into a triangular lattice that persists as a nonequilibrium solid state. Quantitative image analysis identifies crystallization onset when the dimensionless Wigner parameter of Γ —the mean electrostatic-to-thermal energy ratio—reaches ≈ 117 – 149 , comparable to reported critical values for screened electron systems ($\Gamma_c \approx 95$ – 150). Brownian dynamics simulations reproduce the ordering and indicate a Debye screening length of ~ 3 μm under experimental conditions. The colloidal Wigner crystals show reversible ordering under mechanical and magnetic perturbations, providing a simple, real-time platform to study charge-mediated self-organization and the physics of Wigner crystallization under nonequilibrium conditions.

After (Abstract): Elucidating how repulsive interactions evolve to generate ordered structures in nonequilibrium colloidal systems remains a central challenge, owing to the scarcity of experimental platforms that provide particle-resolved, real-time access to their structural evolution. Here we introduce a simple, tunable, room-temperature model in which alginate hydrogel colloids confined within cyclohexyl bromide (CHB) emulsion droplets spontaneously organize under evolving electrostatic interactions. Ba^{2+} ions diffusing from the surrounding aqueous phase progressively crosslink the alginate droplets, increasing their surface charge, while buoyancy compacts them into locally quasi-two-dimensional layers within the CHB phase. As electrostatic repulsion strengthens, the assembly evolves from a disordered state to a triangularly ordered structure. By calibrating Brownian dynamics simulations to the experimentally measured lattice spring constant, we constrain the effective Debye screening length to approximately 2.5 – 3 μm . Quantitative imaging further reveals that ordering emerges once the dimensionless interaction parameter—defined as the ratio of electrostatic to thermal energy—reaches values of approximately 117 – 149 . The ordered state exhibits reversible disordered–order behavior under mechanical and magnetic perturbations, demonstrating a robust nonequilibrium platform for probing charge-regulated colloidal ordering under confinement.

Fully revised Introduction (p3):

Colloidal systems provide a versatile model platform for studying interaction-driven ordering at experimentally accessible length and time scales. Particle-resolved imaging, tunable interactions, and slow relaxation dynamics enable real-time visualization of ordering pathways, offering insights that complement those from atomic and electronic systems.¹⁻⁷ In low-dielectric media such as cyclohexyl bromide (CHB)–decalin mixtures, long-range screened electrostatic repulsion under weak screening can stabilize ordered colloidal phases even at low particle volume fraction.⁸⁻¹⁰ Such systems exhibit rich behaviors, including re-entrant melting, glass formation, and tunable elasticity, which are well described by established theoretical frameworks such as Yukawa, one-component-plasma, and charge-regulation models.^{11, 12} Despite these advances, most prior studies have focused on equilibrium structures, whereas the

[Type here]

time-dependent mechanisms governing surface-charge buildup, evolving screening length, and ordering under confinement have not been extensively explored.

Interaction-driven ordering in colloidal systems generally arises when electrostatic interactions become comparable to or exceed thermal fluctuations, leading to the emergence of ordered structures. In the soft-matter literature, this regime is sometimes described using the term Wigner-type or Wigner-like ordering,⁹⁻¹² reflecting its historical origin in the concept of Wigner crystallization, in which ordered electronic states emerge from the competition between long-range Coulomb interactions and kinetic energy at low carrier densities.¹³⁻¹⁷ In the colloidal assemblies, this terminology is used to denote a regime in which electrostatic interactions dominate over thermal fluctuations and give rise to ordered structures. In the present work, we focus on this interaction-dominated ordering behavior in colloidal particles under screened electrostatic interactions.

Here, we develop a simple, room-temperature platform for studying time-dependent nonequilibrium ordering using soft alginate hydrogel colloids confined within CHB droplets dispersed in an aqueous phase. A microfluidic device produces alginate-nested emulsions with well-controlled droplet size and interfacial curvature, providing reproducible geometric confinement for monitoring ordering pathways. CHB, with its low-dielectric constant, supports strong electrostatic repulsion under weak screening,^{10, 18} while divalent cations (e.g., Ba²⁺) diffusing from the aqueous phase progressively crosslink the alginate droplets, gradually increasing their surface charge. Buoyancy-driven stratification then yields quasi-two-dimensional (2D) layers that spontaneously develop triangular order. As a result of this ion-mediated charging process, the strength of interparticle electrostatic interactions evolves continuously in time. In this context, nonequilibrium refers to the temporal evolution of interparticle interaction strength. Although particle configurations can relax into quasi-steady structural states on relatively short timescales, the interaction strength evolves more slowly, rendering the overall ordering process intrinsically nonequilibrium. We quantify the onset and kinetics of this ordering using particle-resolved imaging, Voronoi/Delaunay analyses, and fast Fourier transform (FFT)-based metrics. To establish a quantitative interaction scale, we calibrate Brownian dynamics (BD) simulations to the experimentally measured lattice spring constant, enabling estimation of the Debye screening length under our experimental conditions. We then characterize the time-dependent strengthening of electrostatic interactions using a dimensionless interaction parameter Γ , defined as the ratio of electrostatic energy to thermal energy. We further map layer-resolved structural ordering and examine the effects of ion valence and ion concentration. Finally, we demonstrate reversible disorder-order transitions under magnetic and mechanical perturbations, together establishing a controllable nonequilibrium platform for studying classical, interaction-driven colloidal ordering.

Revised Fig. 5 (p12):

[Type here]

Fig. 5 Estimation of the Debye screening length and the dimensionless interaction parameter. **a** Bright-field image (i) of an ordered lattice after 12 hr in 100 mM Ba(Ac)₂, used to extract the experimental spring constant k_s^{exp} (Supplementary Movie 1, recorded at 30 fps). Inset (ii) shows the full field of view with the analyzed region (yellow box); inset (iii) shows the probability density of the x -displacement u_x with a Gaussian fit, yielding $k_s^{\text{exp}} \approx 5.02 \text{ nN} \cdot \text{m}^{-1}$. **b** BD-predicted lattice spring constant k_s^{BD} as a function of zeta potential ζ for various κ^{-1} . The horizontal dashed line denotes k_s^{exp} . Error bars are standard deviations over 10 independent BD runs. **c** Contour map of the mismatch $\Delta k_s = |k_s^{\text{BD}} - k_s^{\text{exp}}|$ in the (ζ, κ^{-1}) plane. The white dashed curve marks $\Delta k_s \approx 0$. **d** Dimensionless interaction parameter Γ as a function of incubation time t for different Debye screening lengths κ^{-1} . The pink regions in panels **b–d** highlight the parameter band most consistent with experiment, constraining the Debye length to the few-micrometer range ($\kappa^{-1} \approx 2.5\text{--}3 \mu\text{m}$) and the zeta potential to around $\zeta \approx 50 \text{ mV}$.

Revised section (p14):

Dimensionless interaction parameter and time-dependent ordering

We define a dimensionless interaction parameter Γ to quantify the relative strength of electrostatic interactions compared to thermal fluctuations in our colloidal system dispersed in CHB. Interparticle repulsion is modeled using the Yukawa (screened Coulomb) potential

$E_{\text{pot}} = \frac{4\pi\epsilon_0\epsilon\zeta^2a^2e^{2\kappa a}}{re^{\kappa r}}$, where a is the particle radius, r is the interparticle separation, ϵ_0 is the vacuum permittivity, and κ^{-1} is the Debye screening length. Although the alginate particles are mechanically soft, their interactions in CHB are dominated by the charged surface layer

[Type here]

and occur at separations far larger than those at which mechanical deformation would be relevant. The Yukawa potential therefore provides an appropriate effective pair interaction for the present system. Using the thermal energy $k_B T$ as the reference energy scale, the interaction parameter is defined as $\Gamma = \frac{E_{\text{pot}}}{k_B T} = \frac{\pi \epsilon_0 \epsilon \zeta^2 \langle D \rangle^2 e^{\kappa \langle D \rangle}}{k_B T \langle r_{\text{cc}} \rangle e^{\kappa \langle r_{\text{cc}} \rangle}}$, using experimentally measured values of $\langle D \rangle$, $\langle r_{\text{cc}} \rangle$, and ζ (Fig. 2e and Supplementary Fig. 2). Low Γ corresponds to weakly correlated, fluid-like configurations, whereas large Γ indicates increasingly interaction-dominated, ordered colloidal states.

Fig. 5d shows that Γ increases with incubation time and with increasing screening length, reflecting the progressive strengthening of electrostatic repulsions. A slight dip in Γ at shorter screening lengths can be attributed to the lower $\langle r_{\text{cc}} \rangle$ values observed in the disordered state. Considering the range of $\kappa^{-1} \approx 2.5\text{--}3 \mu\text{m}$ estimated from the BD analysis (Fig. 5b,c), Γ reaches values of $\sim 117\text{--}149$ at $t \approx 6$ hr, coinciding with the experimentally observed onset of hexagonal ordering (Fig. 2). At later times ($t \gtrsim 8$ hr), Γ exceeds ~ 220 (pink regions in Fig. 5d), corresponding to a strongly interaction-dominated regime in which robust colloidal ordering is established. Although the microscopic mechanisms underlying electronic Wigner crystallization are fundamentally different from those governing colloidal ordering, the interaction parameter values at which stable ordering emerges in our colloidal system are of similar numerical magnitude to values often quoted for screened electronic Wigner systems ($\sim 95\text{--}150$).^{14-16,39} Here, this numerical correspondence is not interpreted as evidence of physical equivalence, but rather as providing a useful reference point for identifying the boundary between weakly correlated and strongly ordered regimes in colloidal assembly systems.

Revised Discussion (p 17):

The alginate–CHB platform developed here provides a controlled route to study interaction-driven ordering in a nonequilibrium colloidal environment. As Ba^{2+} ions diffuse into the CHB phase and progressively crosslink the alginate droplets, surface charge builds up, strengthening the screened electrostatic repulsion that drives the transition from a disordered state to an ordered structure. By calibrating BD simulations to the experimentally measured lattice spring constant, we estimate the Debye screening length as $\kappa^{-1} \approx 2.5\text{--}3 \mu\text{m}$ in the alginate colloid system. This estimate remains robust when particle-size polydispersity is incorporated into the simulations, indicating that the spring constant primarily reflects the underlying interaction scale rather than size heterogeneity. The inferred screening length also falls within the range reported previously for colloidal systems in low-dielectric media.^{18, 33-36}

Building on this interaction length scale, we use the dimensionless interaction parameter Γ to summarize the time-dependent strengthening of electrostatic interactions during ordering. As surface charge builds up through ion diffusion and alginate hydrogelation, the resulting electrostatic repulsion strengthens, leading to an increase in Γ with incubation time. Γ reaches values of $\sim 117\text{--}149$ at $t \approx 6$ hr for $\kappa^{-1} \approx 2.5\text{--}3 \mu\text{m}$, coinciding with the experimentally observed onset of triangular ordering. At later times ($t \gtrsim 8$ hr), Γ exceeds ~ 220 , corresponding to a strongly interaction-dominated regime in which robust colloidal ordering is established. At each incubation time, particle configurations appear to relax on timescales much shorter than those associated with surface-charge buildup and screening evolution, allowing the system to reach quasi-steady structural states despite the continuous change in interaction strength.

[Type here]

This separation of timescales justifies the use of equilibrium-based structural descriptors—such as the lattice spring constant and F —at each stage, even though the overall ordering process remains intrinsically nonequilibrium.

Beyond providing quantitative insights into interaction-driven ordering, the alginate–CHB platform offers practical advantages for studying soft-matter systems under weakly ionic conditions. Stable ordered structures arise from simple and accessible steps—sonication and incubation in contact with a Ba^{2+} -containing solution—without requiring complex particle synthesis. Although complete reproducibility is limited by the inherently nonequilibrium nature of the system, the platform remains a convenient and adaptable testbed for time-resolved studies under varied conditions, owing to its electrostatic tunability and compatibility with real-time optical imaging. Looking ahead, several directions could further deepen the mechanistic understanding of the observed ordering. Incorporating single-particle dynamical analyses, such as mean-squared displacements and cage escape times, would provide direct insight into relaxation pathways beyond the structural evolution emphasized here. Microscale temperature control could enable systematic exploration of thermal stability, while dilute-condition direct-force measurements between alginate hydrogels—such as optical tweezer–based interaction profiling^{4,18}—could further refine interaction models and clarify melting pathways. In addition, although Ba^{2+} ions clearly penetrate the CHB medium to crosslink alginate droplets, the microscopic mechanisms governing this ion transfer remain incompletely understood. Future studies quantifying interfacial partitioning, reaction–diffusion kinetics, and counterion exchange will be essential for determining how charge buildup and screening evolve in weakly dissociative media. Together, these efforts would provide a deeper mechanistic foundation for electrostatically mediated colloidal ordering under nonequilibrium conditions and broaden the framework for understanding charge-regulated assembly in low-dielectric environments.

Other comments

1. The term “nonequilibrium” is used frequently. This could be justified if single-particle dynamical analyses were included, which are often the primary strength of colloidal systems. At present, the analysis focuses on the static structure at different time points. It would strengthen the claim to discuss or provide data on particle-level dynamics. For example, how is the “quasi-steady state” achieved? Or are these metastable? Is the dynamics fully arrested at 12 hours? Are structural relaxation times long compared to the observation window, yet short compared to the timescale of charge buildup? Addressing these points would also help justify the use of equilibrium-based analyses.

We thank the reviewer for this comment. In the revised manuscript, we now explicitly define “nonequilibrium” as arising from the continuous temporal evolution of interparticle interactions, driven by gradual surface-charge buildup and changes in electrostatic screening, rather than from persistent dynamical agitation or complete kinetic arrest. We clarify in the Introduction and Discussion that particle configurations appear to relax into quasi-steady structural states on shorter timescales, whereas the interaction strength evolves more slowly, providing a clear timescale separation. This explains why equilibrium-based structural descriptors can be meaningfully applied at each stage despite the intrinsically nonequilibrium nature of the overall ordering process.

[Type here]

Before: CHB, with its low-dielectric constant, supports unusually strong electrostatic repulsion between suspended objects, providing a convenient setting for charge-mediated ordering. Crosslinking by divalent-cations (*e.g.*, Ba^{2+}) generated soft, charged alginate particles whose effective repulsion increased over time as surface charge accumulated, while buoyancy drove flotation-induced stratification, yielding locally quasi-2D layers that spontaneously developed triangular lattice.

After (p3): CHB, with its low-dielectric constant, supports strong electrostatic repulsion under weak screening,^{10,18} while divalent cations (*e.g.*, Ba^{2+}) diffusing from the aqueous phase progressively crosslink the alginate droplets, gradually increasing their surface charge. Buoyancy-driven stratification then yields quasi-two-dimensional (2D) layers that spontaneously develop triangular order. As a result of this ion-mediated charging process, the strength of interparticle electrostatic interactions evolves continuously in time. In this context, nonequilibrium refers to the temporal evolution of interparticle interaction strength. Although particle configurations can relax into quasi-steady structural states on relatively short timescales, the interaction strength evolves more slowly, rendering the overall ordering process intrinsically nonequilibrium.

Newly added statements (p18): At each incubation time, particle configurations appear to relax on timescales much shorter than those associated with surface-charge buildup and screening evolution, allowing the system to reach quasi-steady structural states despite the continuous change in interaction strength. This separation of timescales justifies the use of equilibrium-based structural descriptors—such as the lattice spring constant and Γ —at each stage, even though the overall ordering process remains intrinsically nonequilibrium.

Before: Looking ahead, microscale temperature control could probe thermal stability, dilute-condition optical-tweezer measurements of interparticle repulsion could refine the interaction model and clarify melting pathways. Such studies will further bridge colloidal and electronic Wigner crystals and broaden our understanding of charge-driven self-organization in nonequilibrium settings.

After (p18): Looking ahead, several directions could further deepen the mechanistic understanding of the observed ordering. Incorporating single-particle dynamical analyses, such as mean-squared displacements and cage escape times, would provide direct insight into relaxation pathways beyond the structural evolution emphasized here. Microscale temperature control could enable systematic exploration of thermal stability, while dilute-condition direct-force measurements between alginate hydrogels—such as optical tweezer-based interaction profiling^{4, 18}—could further refine interaction models and clarify melting pathways. In addition, although Ba^{2+} ions clearly penetrate the CHB medium to crosslink alginate droplets, the microscopic mechanisms governing this ion transfer remain incompletely understood. Future studies quantifying interfacial partitioning, reaction-diffusion kinetics, and counterion exchange will be essential for determining how charge buildup and screening evolve in weakly dissociative media. Together, these efforts would provide a deeper mechanistic foundation for

[Type here]

electrostatically mediated colloidal ordering under nonequilibrium conditions and broaden the framework for understanding charge-regulated assembly in low-dielectric environments.

2. The reported size polydispersity of 15–20 % is quite high and could frustrate crystallization. How is this polydispersity estimated? Is it calculated across different planes, and if so, is that appropriate given that size-dependent stratification occurs?

The reported particle-size polydispersity was estimated layer-by-layer from optical microscopy image analysis, rather than by pooling particles across different planes. Specifically, the mean diameter and standard deviation were extracted for the analyzed layer at each time point (Fig. 2e) and for each stratified layer (Fig. 3d). Because buoyancy-driven stratification leads to depth-dependent size distributions, reporting layer-resolved statistics is the most appropriate representation of the polydispersity relevant to the observed ordering.

We agree that relatively large polydispersity can frustrate perfect crystallization and increase defect density. To avoid overstatement, we have revised the wording throughout the manuscript to emphasize “ordering” rather than implying ideal crystallization where appropriate.

In addition, to directly assess the impact of polydispersity on microstructure and lattice mechanics, we performed additional BD simulations incorporating experimentally relevant particle-size distributions (Supplementary Note 3 and Supplementary Fig. 9). These simulations reproduce the expected trend that increasing polydispersity degrades long-range order and produces less ordered configurations. Notably, however, the mean lattice spring constant remains largely insensitive to the degree of polydispersity over the range tested, supporting the robustness of our k_s -based interaction-scale calibration in Fig. 5.

Before: Temporal evolution of structural descriptors: mean particle diameter $\langle D \rangle$, mean center-to-center spacing $\langle r_{cc} \rangle$, and 2D packing fraction ϕ .

After (caption of Fig. 2e): Temporal evolution of layer-resolved structural metrics for the top-layer: mean particle diameter $\langle D \rangle$ (with standard deviation obtained from image analysis), mean center-to-center spacing $\langle r_{cc} \rangle$, and 2D packing fraction ϕ .

Before: Layer dependent of $\langle r_{cc} \rangle$, $\langle D \rangle$, and ϕ .

After (caption of Fig. 3d): Layer-resolved structural metrics for each layer: mean center-to-center spacing $\langle r_{cc} \rangle$, mean particle diameter $\langle D \rangle$ (with standard deviation obtained from image analysis), and 2D packing fraction ϕ .

Newly added paragraph (p14): To assess the robustness of the spring-constant-based calibration, we additionally examined how particle-size polydispersity (i.e., the coefficient of variation, CV) influences the simulated lattice mechanics. As detailed in Supplementary Note

[Type here]

3, we introduced experimentally relevant size variations by sampling particle radii from a Gamma distribution^{37,38} and performed BD simulations at matched surface coverage. Increasing polydispersity reproduces the expected melting-like microstructural disorder (Supplementary Fig. 9a–i), consistent with prior studies,^{37,38} yet the resulting lattice spring constant remains largely insensitive to CV over the range tested (Supplementary Fig. 9j). This insensitivity indicates that the lattice spring constant predominantly reflects the underlying interaction scale rather than the degree of polydispersity, supporting the reliability of the k_s -based screening-length calibration used in Fig. 5b,c.

Before: The same screening length ($\kappa^{-1} \approx 3 \mu\text{m}$) also produced the best agreement between simulated and experimental lattice spring constants. While our BD simulations did not explicitly include interaction heterogeneity or polydispersity—both known to promote melting in colloids^{47,48}—the consistent estimate of $\kappa^{-1} \approx 3 \mu\text{m}$ within the experimentally relevant $\zeta \sim 50\text{--}60 \text{ mV}$ range indicates that finite electrostatic screening is robust in this nonequilibrium environment.

After (p17): By calibrating BD simulations to the experimentally measured lattice spring constant, we estimate the Debye screening length as $\kappa^{-1} \approx 2.5\text{--}3 \mu\text{m}$ in the alginate colloid system. This estimate remains robust when particle-size polydispersity is incorporated into the simulations, indicating that the spring constant primarily reflects the underlying interaction scale rather than size heterogeneity. The inferred screening length also falls within the range reported previously for colloidal systems in low-dielectric media.^{18,33–36}

Newly added Supplementary Fig. 9 (pS10 in SI):

[Type here]

Supplementary Fig. 9 Effects of particle-size polydispersity on microstructure and lattice spring constant in BD simulations. **a-d** Delaunay triangulations of simulated colloidal lattices for varying particle-size polydispersity, quantified by the coefficient of variation (CV): $CV = 0$ (a), 0.13 (b), 0.2 (c), 0.3 (d). Colors indicate distinct coordination environments. **e-h** Corresponding histograms of the coordination number for each CV condition, showing broadening distributions as polydispersity increases. **i** Radial distribution function (rdf) plotted as a function of normalized separation $r \cdot \langle R \rangle^{-1}$, illustrating the progressive attenuation of peak amplitudes and the loss of long-range order as CV increases. **j** Mean lattice spring constant (k_s^{BD}) extracted from BD simulations as a function of CV , showing that k_s^{BD} remains largely insensitive to particle-size polydispersity within the range tested. Error bars represent the standard deviation over ten independent simulation runs.

Newly added Supplementary Note 3 (pS10 in SI):

Supplementary Note 3. Implementation of particle-size polydispersity in BD simulations

To examine how particle-size polydispersity affects the microstructure and the effective lattice spring constant (k_s) of the colloidal assembly structure, we performed BD simulations with $N = 100$ particles interacting via a screened-Coulomb (Yukawa) potential. The box size was chosen such that the surface coverage matched that used in Fig. 5a-c. Experimentally, the top-layer alginate hydrogel particles in Fig. 5a exhibit a mean radius $\langle R \rangle_{\text{exp}} \approx 1.42 \mu\text{m}$ and a standard deviation $\sigma_{R,\text{exp}} \approx 0.19 \mu\text{m}$, corresponding to a coefficient of variation, $CV_{\text{exp}} = \sigma_{R,\text{exp}}/\langle R \rangle_{\text{exp}} \approx 0.13$. To implement controlled polydispersity while ensuring strictly positive radii, we sampled particle sizes from a Gamma distribution, $R \sim \Gamma(k, \theta)$,^{3,4} where the shape parameter k and scale parameter θ satisfy a mean $\mu = \langle R \rangle = k\theta$ and a variance $\sigma^2 = k\theta^2$. The coefficient of variation (CV) is $CV = \frac{\sigma}{\mu} = \frac{1}{\sqrt{k}}$, which gives $k = \frac{1}{CV^2}$ and $\theta = CV^2\mu$. In simulations, we set the target mean radius to $R_0 = 1.42 \mu\text{m}$ and varied the imposed CV . We then sampled $R_i \sim \Gamma(k, \theta)$, $i = 1, \dots, N$.

Particles were initialized on a slightly perturbed square lattice in a periodic box. Each particle experienced overdamped Brownian dynamics with drag coefficient $\gamma_i = 6\pi\eta R_i$ and diffusion coefficient $D_i = \frac{k_B T}{\gamma_i}$. The pairwise Yukawa interaction between particles i and j at separation r_{ij} had effective amplitude, $U_{0,ij} = 4\pi\epsilon_0\epsilon\zeta^2\kappa R_i R_j e^{\kappa(R_i+R_j)}$, and force $F_{ij}(r_{ij}) = U_{0,ij} \frac{e^{-\kappa r_{ij}}}{\kappa r_{ij}^2} (1 + \kappa r_{ij})$. After an equilibration period, particle trajectories were recorded at fixed time intervals.

As shown in Supplementary Fig. 9a–h, Delaunay triangulations and coordination-number histograms show that increasing CV progressively reduces the fraction of sixfold-coordinated sites, giving rise to defect-rich, melted-like configurations. This trend is consistent with our earlier Monte Carlo results, demonstrating that interaction heterogeneity promotes melting and destabilizes long-range order.^{3,4} Likewise, the radial distribution function (rdf) becomes increasingly attenuated and broadened with larger CV , reflecting a clear loss of long-

[Type here]

range order (Supplementary Fig. 9i). In contrast to the strong microstructural degradation, the effective lattice spring constant k_s shows virtually no dependence on CV across the range studied. The mean values from ten independent runs overlap substantially (Supplementary Fig. 9j), indicating that the averaged local confining stiffness is relatively insensitive to particle-size polydispersity despite pronounced structural disorder. These results support the use of k_s as a reliable measure of the interaction scale in our colloidal assemblies and validate the interpretation of Fig. 5 in the main text.

3. The zeta potential (ζ) is a crucial parameter for the quantitative claims, including the calculation of Γ_c and the BD simulations. The value is derived from a proxy measurement in a bulk “vial geometry,” which may not accurately reflect the electrochemical environment inside the confined CHB droplet. The ion concentration, interfacial curvature, and transport dynamics are likely very different. The authors should discuss this limitation and/or provide a stronger justification for why the bulk measurement is considered a reasonable approximation for the droplet system.

We thank the reviewer for raising this important point. We fully agree that the zeta potential measured in the bulk vial geometry does not identically reproduce the electrochemical environment inside a confined CHB droplet, where ion concentration, interfacial curvature, and transport dynamics may differ. For this reason, we do not interpret the measured ζ as an exact local value within the droplet, but rather as an effective proxy for the evolving interaction strength in the CHB phase.

The use of the bulk ζ measurement is justified by several considerations. First, in both the vial and droplet geometries, alginate particles are dispersed in the same CHB medium and acquire charge through the same physical mechanism— Ba^{2+} ions diffusing from the aqueous phase and binding to alginate. Thus, while the geometry differs, the dominant charge-generation pathway is shared. Second, the time-dependent increase in ζ measured in the vial geometry occurs on a similar timescale to the onset of ordering in both the confined droplet and the flattened sessile-drop configurations (Supplementary Fig. 10). Notably, the sessile drop geometry closely resembles the vial configuration in that the CHB phase is in contact with an extended aqueous reservoir, making it a particularly relevant intermediate system. This correspondence provides experimental evidence that the bulk ζ measurement captures the dominant charging dynamics governing ordering in the droplet system.

We have added these points in the revised manuscript.

Before: The magnitude of zeta potential ζ increased with incubation time, indicating progressive surface-charge buildup from Ba^{2+} -carboxylate interactions and continued crosslinking, consistent with the observed crystallization kinetics (Fig. 1d).

After (p6): The magnitude of zeta potential ζ increased with incubation time, indicating progressive surface-charge buildup from Ba^{2+} -carboxylate interactions and continued

[Type here]

crosslinking,²⁸ consistent with the observed ordering kinetics (Fig. 1d). Although the vial geometry does not replicate the confined droplet environment exactly, it captures the same Ba^{2+} -mediated charging mechanism in the CHB phase and thus provides a reasonable effective measure of the evolving interaction strength.

Before: Using the Yukawa potential (Supplementary Fig. 8) and experimental parameter values, BD yields $k_s^{\text{BD}}(\zeta, \kappa^{-1})$ (Fig. 5c). The mismatch map $\Delta k_s = |k_s^{\text{BD}} - k_s^{\text{exp}}|$ (Fig. 5d) identifies the best matches ($|\Delta k_s| \approx 0$); that is, the pink-shaded regions in Fig. 5c,d denote the parameter space corresponding to experimentally reasonable zeta potentials of $\sim 50\text{--}60$ mV (Supplementary Fig. 2).

After (p13): Using a Yukawa potential (Supplementary Fig. 8), we performed BD simulations in which both the zeta potential ζ and the screening length κ^{-1} were systematically and independently varied to obtain the lattice spring constant $k_s^{\text{BD}}(\zeta, \kappa^{-1})$ (Fig. 5b). Comparison of this simulated parameter space with the experimentally measured k_s^{exp} yields a mismatch map, $\Delta k_s = |k_s^{\text{BD}} - k_s^{\text{exp}}|$ (Fig. 5c), from which the best agreement ($|\Delta k_s| \approx 0$) is identified. The corresponding region (pink shading in Fig. 5b,c) lies within an experimentally reasonable range of $\zeta \approx 50\text{--}60$ mV (Supplementary Fig. 2) and yields an effective Debye screening length of $\kappa^{-1} \approx 2.5\text{--}3$ μm .

Before: Since this is a nonequilibrium Wigner crystal, it is difficult to assign exact ζ and κ^{-1} values at any given time. However, assuming $\zeta = 50$ mV, we estimate $\kappa^{-1} \approx 3$ μm . Importantly, this prediction overlaps Γ -based estimate in Fig. 5a. It is also consistent with our prior optical tweezer study using poly(methyl methacrylate) (PMMA) particles in a mixture of CHB and n-decane, which reported $\kappa^{-1} \approx 2.5$ μm . Likewise, reported screening lengths in nonpolar or weakly polar media are roughly in the micrometer range—e.g., ~ 1 μm for PMMA particles in cis-decalin/CHB with poly-12-hydroxystearic acid, $\sim 0.5\text{--}12$ μm for polystyrene or PMMA particles in hexadecane with cationic surfactant (sodium di-2-ethyl-hexylsulfosuccinate), and $\sim 0.8\text{--}8$ μm for PMMA in hexane with sorbitan trioleate.

Because the CHB phase is nonequilibrium and weakly ionic, $\kappa^{-1}(t)$ may drift as Ba^{2+} diffuses from the aqueous phase while being consumed by crosslinking. Nevertheless, multiple estimators converge to $\kappa^{-1} \approx 3$ μm , implying a trace but finite ionic strength in CHB. If $\text{Ba}(\text{Ac})_2$ provides the dominant mobile ions in CHB and Na^+ remains bound within the hydrogel network, it is estimated as $C_{\text{Ba}(\text{Ac})_2} = \frac{1}{3} \frac{\epsilon_0 \epsilon k_B T}{e^2 N_A (\kappa^{-1})^2} = 6.91 \times 10^{-10}$ M for $\kappa^{-1} \approx 3$ μm , an extremely dilute concentration that nonetheless suffices to produce the observed screening in CHB. This consistency across Γ -inference and BD-calibration supports our interpretation of the colloidal Wigner-type crystallization under weakly screened conditions.

After (p13): Because the ordering process evolves under conditions of continuously changing interaction strength, assigning unique values of ζ and κ^{-1} at any specific time is inherently difficult. Nevertheless, by assuming a representative $\zeta \approx 50$ mV, the BD–experiment matching yields an effective screening length of $\kappa^{-1} \approx 3$ μm . This estimate is consistent with our prior optical tweezer measurements on poly(methyl methacrylate) (PMMA) particles in CHB/n-

[Type here]

decane mixtures, which reported $\kappa^{-1} \approx 2.5 \mu\text{m}$.¹⁸ More broadly, reported screening lengths in nonpolar or weakly polar media fall in the micrometer range—for example, $\sim 1 \mu\text{m}$ for PMMA particles in cis-decalin/CHB with poly-12-hydroxystearic acid,³³ $\sim 0.5\text{--}12 \mu\text{m}$ for polystyrene or PMMA particles in hexadecane with cationic surfactant (sodium di-2-ethyl-hexylsulfosuccinate),^{34,35} and $\sim 0.8\text{--}8 \mu\text{m}$ for PMMA in hexane with sorbitan trioleate.³⁶

In the present system, the CHB phase remains weakly ionic, and the effective screening length κ^{-1} in CHB evolves in time as Ba^{2+} diffuses from the aqueous phase and is progressively consumed by alginate crosslinking. Despite this gradual evolution, the inferred screening length $\kappa^{-1} \approx 3 \mu\text{m}$ indicates a trace but finite ionic strength in CHB. Assuming that $\text{Ba}(\text{Ac})_2$ provides the dominant mobile ions in CHB while Na^+ remains largely bound within the hydrogel network, the corresponding ion concentration can be estimated as $C_{\text{Ba}(\text{Ac})_2} = \frac{1}{3} \frac{\epsilon_0 \epsilon k_B T}{e^2 N_A (\kappa^{-1})^2} = 6.91 \times 10^{-10} \text{ M}$ for $\kappa^{-1} \approx 3 \mu\text{m}$. This extremely dilute ionic strength is sufficient to account for the observed electrostatic screening in CHB, supporting our interpretation of electrostatically driven colloidal ordering under weakly screened conditions.

Newly added section (p17):

Alginate colloidal ordering in a flattened sessile drop geometry

In addition to the confined CHB droplets discussed above, we verified that alginate colloidal ordering also emerges in a flattened CHB sessile drop geometry with a much larger interfacial radius of curvature (Supplementary Fig. 10a). Despite the altered geometry, the system exhibited a similar time scale for ordering: well-defined hexagonal domains appeared in the CHB phase approximately 8 hr after exposure to Ba^{2+} , consistent with the droplet-based experiments. This demonstrates that the ordering mechanism is robust and not restricted to the confined droplet environment, and that the sessile drop approach effectively illustrates the generality of the phenomenon. These observations also support the use of vial-based ζ -potential measurements to describe the electrostatic charging behavior of alginate hydrogel particles in both geometries.

To clearly visualize the ordering process, we monitored the edge region of the CHB sessile drop, where alginate dispersed structures beneath the CHB/water interface are optically well resolved (Supplementary Fig. 10b,c). Interestingly, the size of alginate droplets exhibited a characteristic transient evolution during the early stage (0–1 hr). The mean droplet size initially increased within the first 10 min and subsequently decreased over the next several hours (Supplementary Fig. 10d). We attribute this behavior to partial coalescence among weakly crosslinked alginate droplets, producing larger and unstable droplets that rise to the interface and merge into the external aqueous phase. Meanwhile, smaller alginate droplets remain in the CHB phase for longer, allowing sufficient contact with diffusing Ba^{2+} ions to build surface charge and form stable hydrogel particles that eventually assemble into ordered colloidal structures.

Newly added section (p23):

Colloidal ordering in a flattened sessile drop

Sodium alginate (1 wt%) was dissolved in DI water and mixed with CHB at a volume ratio of 0.1:100. The mixture was sonicated for 3 min to generate alginate aqueous droplets dispersed

[Type here]

in the CHB phase. Separately, an aqueous phase containing $\text{Ba}(\text{Ac})_2$ and PVA was prepared by mixing 40 mM $\text{Ba}(\text{Ac})_2$ and 2 wt% PVA at a 2:1 volume ratio. A cylindrical chamber, prepared as described above, was further modified to enable the formation of a flattened CHB sessile drop within the aqueous phase. The bottom glass surface was rendered hydrophobic by applying 0.5 wt% OTS in toluene, followed by drying the chamber in a heated oven for several minutes. After surface treatment, the chamber was filled with 2 mL of the $\text{Ba}(\text{Ac})_2/\text{PVA}$ aqueous solution. A few microliters of the alginate-CHB dispersion was then introduced onto the hydrophobic substrate, where the CHB phase spread into a flattened lens-shaped droplet within the aqueous phase (Supplementary Fig. 10a). The three-phase contact angle of this CHB sessile drop could not be measured using a tensiometer (Attension Theta Auto 1B, Biolin Scientific, Sweden), indicating that the CHB/water interface remained effectively flat while still providing geometric confinement for the formation of ordered alginate hydrogel particles within the CHB phase. Importantly, the CHB sessile drop remained localized on the hydrophobic central region and did not reach the adhesive-sealed boundary, ensuring that it never contacted the UV-curable adhesive and eliminating the possibility of impurity generation through adhesive dissolution.¹⁸

Newly added Supplementary Fig. 10 (pS12 in SI):

[Type here]

Supplementary Fig. 10 Alginate colloidal ordering in a flattened CHB sessile drop geometry. **a** Schematic of the sessile drop setup, where alginate droplets in CHB are confined beneath an aqueous Ba^{2+} /PVA layer. **b** Bright-field images showing time-resolved ordering of alginate droplets at the edge region of the sessile drop. **c** Enlarged and contrast-enhanced views of the boxed regions in panel **b**, with FFT insets highlighting the emergence of triangular order. **d** Temporal evolution of alginate droplet size, with an inset showing magnified view of the early-time regime (0–1 hr).

4. The title, “Dynamics of...”, suggests a primary focus on the dynamic evolution and behavior of the particles, and ties back to my question 1 above. In reality, the main emphasis is on time-resolved characterization of structure. While this can be interpreted as a form of dynamics, the absence of particle motion analyses may mislead readers. The authors could either incorporate such analyses or adjust the title and framing to better reflect the focus on structural formation.

We thank the reviewer for this helpful comment. We agree that the original title could give the impression that the primary focus is on detailed particle-level dynamics, whereas the main emphasis of the manuscript is on time-resolved structural evolution driven by changing interparticle interactions. In light of this point, we have revised the title accordingly.

Before: Dynamics of Nonequilibrium Colloidal Wigner-Type Crystals

After (title): Dynamics of Nonequilibrium Colloidal Ordering Driven by Electrostatic Evolution

5. All experimental details, like the rate of image capture, in the manuscript or supplement, would be helpful for the readers.

As the reviewer suggested, we have added additional experimental details to the revised manuscript and the Supplementary Information.

Before: After confirming stable production for ~5–7 min, the CHB droplets were collected into a Petri dish containing the desired ion solution.

After (p20): After confirming stable production for ~5–7 min, the CHB droplets were collected into a cylindrical chamber containing the desired ion solution. The chamber was constructed by bonding a glass ring to a circular coverslip (Marienfeld, No. 1.5H, Germany) using a UV-curable adhesive (Optical Adhesive 81, Norland, USA). The assembled chamber was thoroughly cleaned with ethanol and DI water, followed by oxygen plasma treatment for 30 s using a plasma cleaner (PDC-32G-2, Harrick Plasma, USA). Because the collected CHB droplets are dispersed in the aqueous phase, they do not come into contact with the UV adhesive, thereby eliminating the possibility of impurity generation in CHB through adhesive dissolution.¹⁸

[Type here]

Before: Bright-field videos were acquired with an Olympus DP80 camera at 30 fps.

After (p23): Bright-field videos were acquired at 30 frames per second (fps) using a CCD camera (DP80, Olympus, Japan) mounted on an optical microscope (IX83/FV3000, Olympus, Japan).

Before: Bright-field image (i) of an ordered lattice after 12 hr in 100 mM Ba(Ac)₂, used to extract the experimental spring constant k_s^{exp} (Supplementary Movie 1).

After (caption of Fig. 5a): Bright-field image (i) of an ordered lattice after 12 hr in 100 mM Ba(Ac)₂, used to extract the experimental spring constant k_s^{exp} (Supplementary Movie 1, recorded at 30 fps).

Before: **Supplementary Movie 1:** Video corresponding to Fig. 5b(i), recorded in bright field at 30 fps.

After (pS13 in SI): **Supplementary Movie 1:** Video corresponding to Fig. 5a(i), recorded in bright field at 30 frames per second.

In summary, the manuscript presents strong experimental work and is suitable for publication in a high impact journal such as Nature Communications. However, I recommend a major revision before acceptance, addressing concerns discussed above.

We thank the reviewer again for the positive assessment of our work and for the constructive comments that helped improve the manuscript. We have carefully addressed the concerns raised above and believe that the revised manuscript has been significantly strengthened as a result.

[Type here]

Reviewer #5 (Remarks to the Author):

We thank the reviewer for their contribution to the peer-review process and for participating in the co-review initiative.

REVIEWERS' COMMENTS

Reviewer #1 (Remarks to the Author):

There appears to be several aspects overlooked in the revised version of the manuscript. Therefore, I am not in favor of publication of the manuscript in current form and suggest major revision.

(1) The mechanism of formation of nested droplets is unclear. Are the droplets of sodium alginate already present in CHB prior to it coming in contact with aqueous PVA solution? What drives the nested emulsion formation? kinetics or thermodynamics?

The sodium alginate droplets are already present in CHB prior to contact with the external aqueous PVA solution. This is explicitly stated in both the Results and Methods sections of the original manuscript (p4-5 and p15 in the revised manuscript). Specifically, alginate sub-droplets are first generated inside CHB by sonication (a kinetically driven emulsification step). This pre-formed alginate-in-CHB dispersion is then introduced into the microfluidic device, where CHB droplets containing these alginate sub-droplets are produced in a continuous aqueous PVA phase. Subsequently, these alginate-in-CHB (Alg@CHB) droplets are incubated in a Ba²⁺-containing aqueous solution. During this stage, Ba²⁺ ions from the aqueous phase diffuse across the CHB phase and induce alginate hydrogelation within the internal alginate sub-droplets.

Because this detailed description is already clearly provided in the manuscript, we have not introduced additional modifications to the main text. We respectfully refer the reviewer to these sections for clarification.

(2) I looked through the citation pertaining to explanation for change in zeta potential presented on Page 6, *Biomacromolecules* 2006, 7, 5, 1471–1480. This article does not discuss anything about zeta or charge measurements. Moreover, the cross-linking process is supposed to reduce charge and hence zeta potential is supposed to decrease. Therefore, the explanation provided does not seem correct, although I believe in the data presented.

We thank the reviewer for the careful check. Our citation to *Biomacromolecules* 2006, 7, 1471–1480 was misplaced in the previous version. That article discusses the strong gel formation induced by barium ions, rather than zeta potential or charge measurements. We have now moved this reference to the appropriate section describing the enhanced gel strength associated with Ba²⁺ crosslinking. We apologize for this oversight.

Regarding the reviewer's statement that cross-linking should reduce charge and therefore decrease ζ : we agree that, from a purely stoichiometric viewpoint, divalent-ion crosslinking locally consumes or bridges carboxylate groups (COO⁻) in junction zones, which could reduce the number of "free" charges. However, the experimentally measured zeta potential does not directly report the total number of charged groups inside the hydrogel network. Instead, ζ reflects the effective electrostatic potential at the shear plane, which depends on ion exchange, counterion distribution, and charge density near the particle–medium interface. In our system, Ba²⁺ exposure involves (i) Na⁺→Ba²⁺ exchange and progressive hydrogelation, and (ii) time-

[Type here]

evolving restructuring and compaction of the gel network. These processes can plausibly increase the effective surface charge density and/or modify the interfacial counterion environment, leading to an increase in the measured $|\zeta|$. This interpretation is consistent with our observation that the magnitude of ζ increases with incubation time (Supplementary Fig. 2). It is also supported by the experimental observation that partially gelled droplets at early times exhibit coalescence, whereas fully incubated samples no longer show coalescence (Supplementary Fig. 10), indicating progressive stabilization and strengthening of interparticle electrostatic repulsion over time. We have therefore provided a more detailed explanation in Supplementary Note 1.

Newly added Supplementary Note 1:

Supplementary Note 1. Interpretation of the time-dependent increase in $|\zeta|$ during Ba^{2+} -mediated hydrogelation

The evolution of the zeta potential (ζ) during Ba^{2+} -mediated hydrogelation requires careful interpretation. From a purely stoichiometric perspective, divalent-ion crosslinking bridges carboxylate groups (COO^-) within junction zones, which could reduce the number of free charges. However, the experimentally measured ζ does not directly reflect the total number of charged groups inside the hydrogel network. Instead, ζ represents the effective electrostatic potential at the shear plane and is governed by the interfacial charge density and counterion distribution near the particle–medium interface. In our system, alginate droplets initially contain Na^+ as counterions. Upon exposure to Ba^{2+} , ion exchange ($\text{Na}^+ \rightarrow \text{Ba}^{2+}$) occurs concurrently with progressive hydrogelation. This process alters the counterion environment and the electrostatic structure at the interface. As the gel network matures, restructuring and partial compaction of the hydrogel can modify the effective surface charge density. Consequently, the magnitude of ζ may increase during gelation despite local crosslink formation. Consistent with this interpretation, the measured $|\zeta|$ increases with incubation time (Supplementary Fig. 2). In parallel, partially gelled alginate droplets at early times exhibit coalescence, whereas fully incubated samples no longer show coalescence (Supplementary Fig. 10), indicating progressive stabilization associated with strengthened electrostatic repulsion.

(3) The buoyancy-driven stratification must be substantiated with density measurements or rising velocity calculations

We thank the reviewer for this suggestion. We have included an order-of-magnitude estimate of the rising velocity based on Stokes' law, which yields a migration time of $\sim 10^2$ – 10^4 s (i.e., minutes to a few hours) for droplets in the 0.5–5 μm size range to traverse a 400 μm CHB droplet. This timescale is significantly shorter than the ≈ 6 –8 h required for the emergence of pronounced hexagonal ordering, indicating that buoyancy-driven stratification occurs well before long-range ordering develops. Thus, buoyancy rapidly establishes quasi-2D layering, whereas the delayed onset of hexagonal ordering more likely reflects the time-dependent strengthening of electrostatic interactions during Ba^{2+} -mediated hydrogelation. We have incorporated a brief clarification of this timescale separation in the main text and provided the detailed description in the newly added Supplementary Note 2.

[Type here]

Before: Particles near the top then experienced strong electrostatic repulsion and assembled into locally quasi-2D layers that developed triangular lattice.

After (p5): An order-of-magnitude estimate based on Stokes' law indicates that droplets in the $\approx 0.5\text{--}5\ \mu\text{m}$ size range traverse a $400\ \mu\text{m}$ CHB droplet within $\sim 10^2\text{--}10^4\ \text{s}$ (minutes to a few hours; Supplementary Note 2). Particles near the top therefore form quasi-2D layers on relatively short timescales, providing a geometrically confined environment for subsequent ordering.

Newly added Supplementary Note 2:

Supplementary Note 2. Buoyancy-driven stratification and its timescale

To substantiate the buoyancy-driven stratification, we estimated the rising velocity of alginate aqueous droplets dispersed in CHB using Stokes' law. Taking a representative density difference of $\Delta\rho \approx 320\ \text{kg m}^{-3}$ (CHB $\approx 1.33\ \text{g cm}^{-3}$; 1 wt.% alginate aqueous phase $\approx 1\ \text{g cm}^{-3}$) and a CHB viscosity of $\eta \approx 1.5 \times 10^{-3}\ \text{Pa s}$, the terminal rising velocity is given by $v = \frac{2\Delta\rho g R^2}{9\eta}$, where g is the gravitational acceleration. For droplet diameters in the approximate range $2R \approx 0.5\text{--}5\ \mu\text{m}$, the calculated velocities are on the order of $\sim 0.01\text{--}1\ \mu\text{m s}^{-1}$. The corresponding time required to traverse the full $400\ \mu\text{m}$ CHB droplet is therefore on the order of $\sim 10^2\text{--}10^4\ \text{s}$ (i.e., minutes to a few hours). These estimates indicate that buoyancy-driven stratification occurs on a timescale significantly shorter than the $\approx 6\text{--}8\ \text{h}$ required for the emergence of pronounced hexagonal ordering. This interpretation is consistent with the experimental observations in Fig. 2c, where larger droplets are already concentrated near the upper region of the CHB droplet at early times, prior to the development of long-range hexagonal order. We therefore conclude that stratification and ordering occur on distinct timescales. Buoyancy rapidly produces quasi-2D layering, whereas the delayed onset of hexagonal ordering likely reflects the time-dependent strengthening of electrostatic interactions during Ba^{2+} -mediated hydrogelation.

(4) Figure 4 (a) and (b) - correspondence between particles in (b) identified from analysis of particles in (a) appears poor. Some of the droplets in the images are not in focus, therefore, the image analysis and further calculations need to be checked.

As indicated in the caption, Fig. 4b corresponds to a zoomed-in region taken from the central, in-focus portion. Only particles within this well-focused region were included in the segmentation and subsequent quantitative analysis. We have revised the caption to make this point clearer.

Before: Corresponding segmented, zoomed-in particle maps used for analysis.

After (Caption of Fig. 4b): Corresponding segmented particle maps obtained from the central, in-focus zoomed-in region and used for analysis.

[Type here]

(5) Debye length calculated (Page 13 and Figure 5), which are or the order of few microns, seem unusually high and do not make physical sense. Please consider including reason for this.

The micrometer-scale Debye screening length obtained in our study may appear large in comparison to aqueous systems; however, such values have been well documented in previous studies of colloids dispersed in nonpolar or weakly polar solvents [refs. 18, 33-36 in the main text]. We have already discussed this point in detail in the section “Estimation of Debye screening length”. We therefore respectfully refer the reviewer to that section for clarification.

(6) In Figure 10 (c) in Supplementary, although enlarged and contrast-enhanced images are shown, the emergence of triangular order is not there, contrary to what is claimed. Therefore, the discussion presented in the section "Alginate colloidal ordering in a flattened sessile drop geometry", lacks clarity and brings a question about generality of the observations made in the droplets.

In Supplementary Fig. 10c, clear triangular order is not evident in the earlier time points (i.e., 20 min and 4 h), whereas pronounced triangular ordering becomes clearly visible at the 24 h sample. To clarify this point, we have provided the full-field video of the 24 h sample as Supplementary Movie 2.

Reviewer #2 (Remarks to the Author):

The manuscript presents an experimentally elegant colloidal system that reproduces several key phenomenological features of Wigner crystallization using soft alginate particles in a low-dielectric solvent. The identification of a critical interaction parameter ($\Gamma \approx 120-150$) at the onset of ordering, supported by calibrated Brownian dynamics simulations, is a clear strength and lends quantitative credibility to the analogy.

The newly added analysis of particle-size polydispersity appropriately demonstrates that structural order is sensitive to disorder, while the inferred interaction length scale remains comparatively robust. Nevertheless, the system remains intrinsically nonequilibrium, and the microscopic origin, temporal stability, and reproducibility of electrostatic screening in CHB remain somewhat qualitative. While the separation of timescales provides a reasonable justification for using equilibrium-based structural descriptors, this assumption should be borne in mind when interpreting the results. Overall, the study provides a well-constructed and physically transparent soft-matter analogue of charge-driven crystallization, while also highlighting important directions for future mechanistic refinement (CHB related). The latter could be asked for now or left for future work

We thank the reviewer for the positive and constructive assessment of our work. We are pleased that the quantitative identification of a critical interaction parameter and the polydispersity analysis were viewed as strengths. We agree that the system is intrinsically nonequilibrium and that the microscopic origin and temporal stability of electrostatic screening in CHB merit further investigation. We appreciate the reviewer’s perspective and consider deeper

[Type here]

mechanistic clarification an important direction for future work.

Reviewer #4 (Remarks to the Author):

The revised manuscript by Jung et al. presents a simple and elegant experimental platform to study colloidal ordering driven by electrostatic interactions. The manuscript has improved substantially in clarity and framing, including the title, abstract, and overall narrative. I particularly appreciate the addition of the sessile-drop experiments, which significantly strengthen the robustness and generality of the results.

We sincerely thank the reviewer for the positive assessment and encouraging feedback, and we are pleased that the revisions—particularly the addition of the sessile-drop experiments—have strengthened the clarity and robustness of the manuscript.

I recommend publication in Nature Communications, subject to the following minor comments:

1. The terms triangular and hexagonal ordering are used interchangeably throughout the manuscript. Given that the experiments clearly show dominant sixfold symmetry, I suggest consistently using hexagonal ordering. While triangular motifs underlie a hexagonal lattice, the observed lattice structure is hexagonal, and consistent terminology would improve clarity.

As the reviewer suggested, we have consistently used the term “hexagonal” throughout the revised manuscript.

2. The statement in line 76 (“Although particle ... nonequilibrium”) would benefit from further clarification. It is not immediately clear how structural relaxation could proceed faster than the underlying charge buildup that governs the evolution of interaction strength. This point appears closely related to the statement in line 130 (“the magnitude of the zeta potential ... ordering kinetics,” Fig. 1d), which suggests a coupling between charge buildup and ordering dynamics. A brief clarification reconciling these statements would help improve the physical interpretation.

We agree that clarification is helpful. The apparent ambiguity may arise from the distinction between two different processes: (i) local configurational relaxation, i.e., particle rearrangement at a given interaction strength, and (ii) the global disorder-to-order transition that defines the overall ordering kinetics. In our system, *particle configurations* relax on timescales that are short compared to the gradual buildup of surface charge during Ba²⁺-mediated hydrogelation. At each stage, the particle arrangement rapidly adjusts toward the quasi-steady configuration corresponding to the instantaneous interaction strength. By contrast, the global emergence of long-range hexagonal order is governed by the slower evolution of the electrostatic interaction strength, as reflected by the gradual increase in ζ . In this sense, the system passes through a sequence of quasi-steady structural states while remaining intrinsically nonequilibrium due to the time-dependent interaction strength. We have clarified this point in the revised manuscript.

[Type here]

Before: Although particle configurations can relax into quasi-steady structural states on relatively short timescales, the interaction strength evolves more slowly, rendering the overall ordering process intrinsically nonequilibrium.

After (p4): Although local particle configurations can relax into quasi-steady structural states on relatively short timescales at a given interaction strength, the electrostatic interaction strength itself evolves more slowly due to progressive charge buildup. As a result, the global emergence of long-range hexagonal order is governed by this slower timescale, rendering the overall ordering process intrinsically nonequilibrium.

Comments on 646903_0

The manuscript by Jung *et al.* presents a novel experimental system for studying the formation of two-dimensional ordered colloidal arrays. The approach, involving in-situ crosslinking of alginate hydrogels within low-dielectric CHB droplets, offers an accessible, room-temperature platform for investigating “Wigner-type” crystallization. The experiments are carefully designed, and the resulting structures are thoroughly analyzed, with corroboration from Brownian dynamics simulations. Overall, the manuscript represents a valuable contribution to the field.

My main concern lies with the direct analogy drawn to the electronic Wigner crystal in quantum systems. In my view, this analogy is at best heuristic. As the authors themselves note in the section “*Estimation of Wigner parameter and Debye screening length*”, the quantum phase transition is governed by the competition between long-range Coulomb forces and quantum kinetic energy, whereas the ordering observed here arises from the interplay of screened Yukawa potentials and thermal energy. This is a fundamental distinction that should be clearly articulated early in the Introduction. Even towards the end in the Discussion (starting line 349), the phrasing tends to blur this difference. A systematic study of colloidal Wigner crystallization is important and stands on its own merit; however, the manuscript would benefit from a clear and consistent separation between the colloidal analogue and the quantum electronic case. If the authors intend to maintain a stronger equivalence, some direct experimental evidence would be necessary to justify the claim.

Other comments

1. The term “nonequilibrium” is used frequently. This could be justified if single-particle dynamical analyses were included, which are often the primary strength of colloidal systems. At present, the analysis focuses on the static structure at different time points. It would strengthen the claim to discuss or provide data on particle-level dynamics. For example, how is the “quasi-steady state” achieved? Or are these metastable? Is the dynamics fully arrested at 12 hours? Are structural relaxation times long compared to the observation window, yet short compared to the timescale of charge buildup? Addressing these points would also help justify the use of equilibrium-based analyses.
2. The reported size polydispersity of 15–20 % is quite high and could frustrate crystallization. How is this polydispersity estimated? Is it calculated across different planes, and if so, is that appropriate given that size-dependent stratification occurs?
3. The zeta potential (ζ) is a crucial parameter for the quantitative claims, including the calculation of Γ_c and the BD simulations. The value is derived from a proxy measurement in a bulk “vial geometry,” which may not accurately reflect the electrochemical environment inside the confined CHB droplet. The ion concentration, interfacial curvature, and transport dynamics are likely very different. The authors should discuss this limitation and/or provide a stronger justification for why the bulk measurement is considered a reasonable approximation for the droplet system.
4. The title, “*Dynamics of...*”, suggests a primary focus on the dynamic evolution and behavior of the particles, and ties back to my question 1 above. In reality, the main emphasis is on time-resolved characterization of structure. While this can be interpreted as a form of dynamics, the absence of particle motion analyses may mislead readers. The authors could either incorporate such analyses or adjust the title and framing to better reflect the focus on structural formation.
5. All experimental details, like the rate of image capture, in the manuscript or supplement, would be helpful for the readers.

In summary, the manuscript presents strong experimental work and is suitable for publication in a high-impact journal such as Nature Communications. However, I recommend a major revision before acceptance, addressing concerns discussed above.